# Correcting Errors in Seasonal Arctic Sea Ice Prediction of Earth System Model with Machine Learning

Zikang He[1,2,3], Yiguo Wang[3], Julien Brajard[3], Xidong Wang[1,2], and Zheqi Shen[1,2]

[1]Key Laboratory of Marine Hazards Forecasting, Ministry of Natural Resources, Hohai University, Nanjing, 210098, China

[2]College of Oceanography, Hohai University, Nanjing, 210098, China

[3]Nansen Environmental and Remote Sensing Center and Bjerknes Centre for Climate Research, Jahnebakken 3, Bergen, N-5007, Norway

**Correspondence:** Yiguo Wang (Yiguo.wang@nersc.no) and Xidong Wang (xidong_wang@hhu.edu.cn)

**Abstract.** While Earth system models are essential for seasonal Arctic sea ice prediction, they often exhibit significant errors that are challenging to correct. In this study, we integrate a multilayer perceptron (MLP) machine learning (ML) model into the Norwegian Climate Prediction Model (NorCPM) to improve seasonal sea ice predictions. We compare the online and offline error correction approaches. In the online approach, ML corrects errors in the model's instantaneous state during the model simulation, while in the offline approach, ML post-processes and calibrates predictions after the model simulation. Our results show that the ML models effectively learn and correct dynamical model errors in both approaches, leading to improved predictions of Arctic sea ice during the test period (i.e., 2003-2021). Both approaches yield the most significant improvements in the marginal ice zone, where error reductions in sea ice concentration exceed 20%. These improvements vary seasonally, with the most substantial enhancements occurring in the Atlantic, Siberian, and Pacific regions from September to January. The offline error correction approach consistently outperforms the online error correction approach. This is primarily because the online approach targets only instantaneous model errors on the 15th of each month, while errors can grow during the subsequent one-month model integration due to interactions among the model components, damping the error correction in monthly averages. Notably, in September, the online approach reduces the error of the Pan-Arctic sea ice extent by 50%, while the offline approach achieves a 75% error reduction.

# 1 Introduction

According to satellite observations, the Arctic sea ice extent (SIE) rapidly declines throughout all calendar months during the recent decades (e.g., Serreze et al., 2007; Onarheim et al., 2018; Wang et al., 2022; Heuzé and Jahn, 2024). The most significant reductions occurred in the summer and autumn (e.g., September, Stroeve et al., 2014). The wider open ocean leads to growing socioeconomic activities in the Arctic (e.g., fisheries, shipping, and resource extraction). These increased human activities highly demand accurate seasonal predictions of Arctic sea ice conditions (Jung et al., 2016; Wagner et al., 2020). The Sea Ice Outlook, managed by the Sea Ice Prediction Network, produces monthly reports during the Arctic sea ice retreat season. These monthly reports synthesize input from the international research community devoted to enhancing sea ice predictions. Recently, Bushuk et al. (2024) evaluated 17 statistical models, 17 dynamical models, and 1 heuristic approach in predicting September Arctic sea ice. They found that dynamical and statistical models are overall comparable in predicting the Pan-Arctic SIE, and dynamical models generally outperform statistical models in predicting the regional SIE and sea ice concentration (SIC, i.e., local quantities). Bushuk et al. (2024) also suggested that the dynamical models must further improve their initialization and model resolution to reduce prediction errors.

Data assimilation (DA) integrates observations with dynamical models to optimally estimate the state of the climate system (Penny and Hamill, 2017; Carrassi et al., 2018). It has widespread application in producing reanalysis (Saha et al., 2006; Dee et al., 2011; Laloyaux et al., 2018; Zuo et al., 2019; Hersbach et al., 2020), offering comprehensive, continuous, and dynamically consistent reconstructions of past climate states. Simultaneously, many prediction centers are transitioning to use DA methods to mitigate uncertainties in initial conditions (Wang et al., 2013; Vitart et al., 2017; Blockley and Peterson, 2018; Kimmritz et al., 2019; Wang et al., 2019; Bushuk et al., 2024). The improved quantity and quality of observations across different climate system components and advanced DA methods enable more precise initial conditions for seasonal predictions of Arctic sea ice. Nevertheless, even with perfect initial conditions, prediction errors escalate over time due to the inherent deficiencies of dynamical models in emulating the true climate system (gray and green lines in Figure 1). This underscores the necessity for dealing with prediction errors.

Machine learning (ML) has recently emerged as a data-driven technique to mitigate dynamical prediction errors. Two prevalent approaches include constructing an ML-dynamical hybrid model (e.g., Brajard et al., 2021; Watt-Meyer et al., 2021) and post-processing/calibrating model outputs (e.g., Yang et al., 2023; Palerme et al., 2024). The former is considered as online error correction, while the latter refers to offline error correction.

In the context of the online error correction, ML is applied to correct errors in the instantaneous model state (i.e., initial conditions for the following model integration) and sequentially applied to update the instantaneous model state during simulation (e.g., Brajard et al., 2021), referring to an ML-dynamical hybrid model (purple line in Figure 1). Such online error correction approaches have been investigated in both an idealized framework (e.g., Watson, 2019; Brajard et al., 2021) and real applications (e.g., Watt-Meyer et al., 2021).

Watson (2019) examined the tendency error correction approach in the Lorenz 96 model. Brajard et al. (2021) explored the resolvent error correction approach in the two-scale Lorenz model as well as in a low-order coupled ocean-atmosphere

model called the Modular Arbitrary-Order Ocean-Atmosphere Model (MAOOAM, De Cruz et al., 2016). Watt-Meyer et al. (2021) demonstrated that the online error correction can improve the short-term forecasting skill and accuracy of precipitation simulation while the dynamical model can run indefinitely without numerical instabilities arising. Gregory et al. (2024) applied ML to correct sea ice errors in an ocean-ice coupled model and demonstrated that ML can effectively reduce sea ice bias in a 5-year simulation. So far, the ML-based online error correction method has not been tested for seasonal sea ice prediction in an Earth system model. In this study, we build and assess a hybrid model combining ML and a state-of-the-art Earth system model for seasonal prediction of Arctic sea ice.

On the other hand, the offline error correction consists in performing post-processing (also called calibration) of the dynamical model predictions (blue line in Figure 1). ML is trained to predict errors for time-averaged model outputs (e.g., daily or monthly outputs) and applied to correct errors present in raw predictions. The most common error correction methods employed in sea ice prediction (Bushuk et al., 2024) are relatively simple (e.g., correction of the mean error or a linear regression adjustment, Blanchard-Wrigglesworth et al., 2017). More recently, Palerme et al. (2024) applied ML to improve the skill of sea ice forecasts on the weather timescale. Overall, they illustrated that ML-based offline calibration reduced the SIC prediction errors by $41\%$ and the ice edge distance error by $44\%$. Their application is mainly focused on short-term sea ice prediction within 10 days in an ocean-ice coupled model. In this study, we apply and assess the ML-based calibration for seasonal prediction of Arctic sea ice in a state-of-the-art fully-coupled Earth system model.

In this study, we apply ML to the Norwegian Climate Prediction Model (NorCPM, Wang et al., 2019), a fully-coupled Earth system model, for seasonal prediction of Arctic sea ice. We test and compare the ML-based online and offline error correction approaches. In the online approach, we build a hybrid model combining ML and NorCPM to update the instantaneous sea ice state during the production of seasonal predictions. In the offline approach, we use ML to calibrate raw seasonal predictions of Arctic sea ice. The comparison between the two approaches within the same framework delivers new insights for the sea ice prediction community into how to effectively use ML for seasonal Arctic sea ice predictions.

The paper is organized as follows: section 2 presents the dynamical model, data, ML-based error correction approaches, experimental design, and metrics for validation. Section 3 shows the results of different experiments. We finish with discussions and conclusions in section 4.

## 2 Data and Methods

### 2.1 Norwegian Climate Prediction Model

The dynamical model we used is NorCPM (Counillon et al., 2014, 2016; Wang et al., 2016, 2017; Kimmritz et al., 2018, 2019). It combines the Norwegian Earth System Model version 1 (NorESM1, Bentsen et al., 2013) and a deterministic formulation of an advanced flow-dependent DA method named ensemble Kalman filter (EnKF, Sakov and Oke, 2008).

NorESM1 (Bentsen et al., 2013) is a fully-coupled Earth system model used for climate simulations. Its ocean component is the Bergen Layered Ocean Model (BLOM, Bentsen et al., 2013) – an updated version of the isopycnal coordinate ocean model MICOM (Bleck et al., 1995). The sea ice component is the Los Alamos sea ice model version 4 (CICE4, Gent et al.,

2011; Holland et al., 2012). The atmospheric component is a variant of the Community Atmosphere Model version 4 (CAM4-Oslo, Kirkevåg et al., 2018). The land component is the Community Land Model (CLM4, Thornton, 2010; Lawrence et al., 2011). Furthermore, the version 7 coupler (CPL7, Craig et al., 2012) is utilized for inter-component communication and interaction. The external forcings follow the protocol of the Coupled Model Intercomparison Project Phase 5 (CMIP5) historical experiment (Taylor et al., 2012).

The atmospheric and land components are situated on the National Center for Atmospheric Research (NCAR) finite-volume $2°$ grid, featuring a regular $1.9° \times 2.5°$ latitude–longitude resolution with 26 hybrid sigma–pressure levels extending to 3 hPa. The ocean and sea ice components utilize NCAR's gx1v6 horizontal grid, which is a nominal $2°$ resolution curvilinear grid with the northern pole singularity shifted over Greenland (Bethke et al., 2021). This grid is enhanced both meridionally towards the equator and zonally and meridionally towards the poles. The ocean component comprises 51 isopycnic layers, featuring a bulk mixed layer representation on top with two layers having time-evolving thicknesses and densities.

The sea ice component is equipped with five ice thickness categories to account for the different thermodynamic and dynamic properties of ice with different thicknesses. The volume of snow and ice, energy content, as well as SIC, surface temperature, and the volume-weighted mean ice age are determined for each of the ice thickness categories (Bentsen et al., 2013; Kimmritz et al., 2018, 2019).

NorESM1 tends to overly produce thick sea ice, especially in the polar oceans adjacent to the Eurasian continent. This is partly due to factors such as weaker winds across the polar basin and overestimated Arctic cloudiness, which leads to little summer snowmelt. Consequently, the summer SIE in the Arctic has large positive biases, contributing to an underestimation of global temperatures (Bentsen et al., 2013; Bethke et al., 2021).

NorCPM uses the EnKF to update unobserved ocean and sea ice variables by leveraging state-dependent covariance from the simulation ensemble (Kimmritz et al., 2018, 2019). The EnKF allows the assimilation of observations of various types while accounting for observational errors, spatial coverage, and the evolving covariance with the climate state. The EnKF accounts for uncertainties in initial conditions to generate ensemble predictions, which evolve in time and provide time- and space-dependent error estimates.

NorCPM employs anomaly-field assimilation (Kimmritz et al., 2019; Wang et al., 2019; Bethke et al., 2021) in which the climatology of the observations is replaced by the model climatology calculated from the ensemble mean of the model historical simulation (without assimilation). While the anomaly-field assimilation keeps the model close to its attractor and helps to reduce the model drift during the monthly model integration (Carrassi et al., 2014; Weber et al., 2015), it does not significantly change model biases.

## 2.2 Data

In this study, we use the reanalysis of NorCPM as the "truth" to assess the improvement achieved by the ML-based error correction approaches. First, it is because NorCPM performs anomaly-field assimilation. The large model biases are not corrected by DA (section 2.1) and thus the analysis increment of the reanalysis used to build the online error correction model (section 2.3) does not take into account model biases. Second, the online error correction approach needs to consistently update SIC

in each category, sea surface temperature (SST), and sea surface salinity (SSS) under sea ice, which are often not observed. The reanalysis of NorCPM is a physically consistent construction of the Earth system (Counillon et al., 2016; Kimmritz et al., 2019) and provides a reasonable and physically consistent estimation of these variables. Finally, the reanalysis combining observations with NorESM represents the upper limit of the sea ice predictability of NorCPM.

The reanalysis is available from 1980 to 2021 with 30 ensemble members. The initial states of the reanalysis on 15 January 1980 are taken from a NorESM ensemble run integrated from 1850 to 1980 with CMIP5 historical forcings. In this reanalysis, NorCPM assimilates monthly anomalies of SST, SIC, and subsurface hydrographic profile data in the middle of each month.

From 1980 to 2002, the climatology used for anomaly-field assimilation is defined over the period 1980–2010. SST and SIC observations are from HadISST2 (Titchner and Rayner, 2014) and subsurface hydrographic profile data from EN4.2.1 (Good
et al., 2013). The assimilation process contains two steps addressed in Kimmritz et al. (2019): firstly, hydrographic DA updates the ocean state (Wang et al., 2017). Subsequently, SST and SIC DA occur and update the sea ice and ocean states within the ocean mixed layer. From 2003 to 2021, the climatology utilized for anomaly-field assimilation is defined from 1982 to 2016. SST and SIC observations are from OISST (Reynolds et al., 2007) and subsurface hydrographic profile data from EN4.2.1 (Good et al., 2013). Strong-coupled DA is performed to simultaneously update the sea ice and ocean states in a single
step.

After each assimilation step, a post-processing step is used to ensure the physical consistency of state variables. For example, the volume of each sea ice category is proportionally adjusted based on the updated SIC (Kimmritz et al., 2018, 2019). The other model components, such as the atmosphere and land, are dynamically adjusted through the coupler during model integration between two assimilation steps.

**2.3   Online error correction approach**

The online error correction approach is built from the analysis increment of the reanalysis introduced in section 2.2 (Brajard et al., 2021; Gregory et al., 2024) and sequentially applied to update the instantaneous model state in the middle of each month during prediction simulation (purple line in Figure 1), which is similar to the reanalysis system (section 2.2).

The monthly model integration of the reanalysis (section 2.2) can be described as follows:

$$\mathbf{x}_k^f = \mathcal{M}(\mathbf{x}_{k-1}^a), \tag{1}$$

where $\mathbf{x}_k^f$ represents the forecasted instantaneous model state at $t_k$, $\mathcal{M}$ represents the dynamical model integration from time $t_{k-1}$ to $t_k$ (section 2.1). During the analysis, DA uses available observations to generate $\mathbf{x}_k^a$ — an updated instantaneous model state and initial conditions for the next monthly model integration from time $t_k$ to time $t_{k+1}$.

The online approach is to emulate the difference between the forecast and the analysis $\mathbf{x}_k^f - \mathbf{x}_k^a$, which corresponds to the
opposite of the analysis increment in DA. The error prediction model can be expressed as:

$$\varepsilon = \mathcal{M}_e(\mathbf{x}^f), \tag{2}$$

where $\mathcal{M}_e$ represents the data-driven model taking the instantaneous model state $\mathbf{x}^f$ as input and $\varepsilon$ represents the predicted model error.

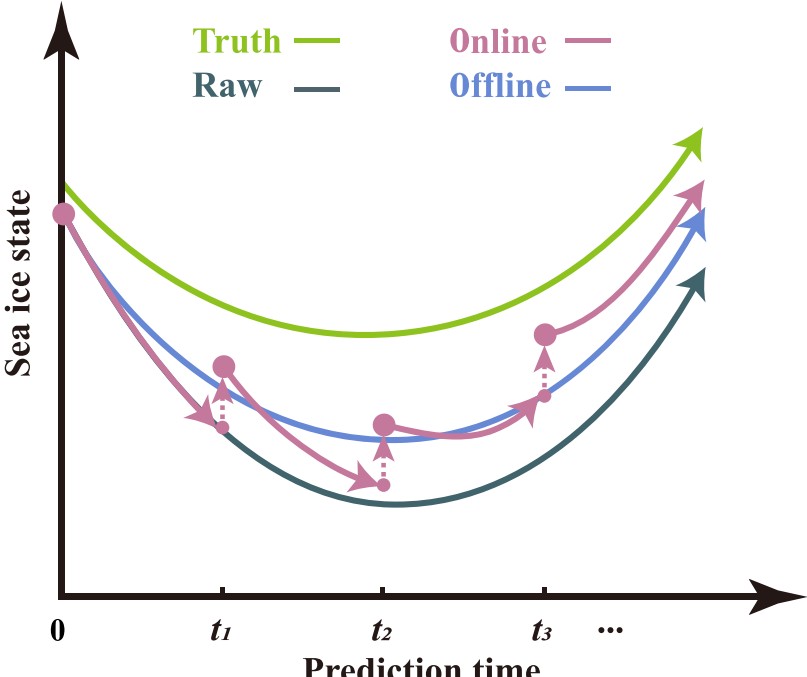

**Figure 1.** Schema for the online and offline ML-based error correction approaches. The green line represents the "truth". The gray line represents dynamical prediction without error correction. The purple (blue) line represents prediction with online (offline) ML-based error correction. The purple dashed arrows indicate pauses during the prediction production, facilitating correction to the instantaneous model states.

The hybrid model, incorporating the dynamic model and the online error correction model, can be expressed as follows:

$$\mathbf{x}_l^h = \mathcal{M}(\mathbf{x}_{l-1}^h) - \mathcal{M}_{\mathrm{e}}(\mathcal{M}(\mathbf{x}_{l-1}^h)), \tag{3}$$

where $\mathbf{x}_l^h$ represents the error-corrected instantaneous model state at $t_l$ during the prediction.

We aim to correct SIC, SST, and SSS errors in the ice-covered area, which are directly associated with the sea ice condition. Considering the seasonality of the error of the sea ice state, we build one error correction model for each calendar month. Also, we employ a running training strategy and use the most recent 11 years of data before the prediction month (the first 10 years for training and the last year for validation). The input feature contains latitude, SST, SSS, five categories of SIC, and five categories of sea ice volume in the middle of the month. The output feature consists of errors in SST, SSS, and 5 categories of SIC (Table 1). Please refer to section 2.5 for the ML configuration.

Before restarting the model after applying online error correction, it is essential to ensure that the updated variables remain within physical limits (e.g., SIC between $0\%$ and $100\%$) and maintain consistency with non-updated variables. If unphysical values or inconsistencies arise, they can lead to model instability. To prevent these issues, we apply a post-processing method specifically designed for NorCPM (Kimmritz et al., 2018):

**Table 1.** Information about online and offline ML-based error correction models

|  | Online ML-based model | Offline ML-based model |
|---|---|---|
| Input features | Instantaneous SST, SSS, latitude, 5 categories SIC, and sea ice volume | Monthly SST, SSS, latitude, SIC, and sea ice volume |
| Output features | Instantaneous SST, SSS, and 5 categories SIC errors | Monthly SIC prediction error |
| Data | The most recent eleven years data (ten years for training and one year for validation) | |
| Remark | Only apply to sea-ice covered grids in the Arctic with SIC values greater than 1%. | |

- If SIC in any thickness category falls below 0% or exceeds 100%, it is set to 0% or 100%, respectively.

- If the total SIC across all thickness categories exceeds 100%, SIC values in each category are proportionally scaled to ensure the total does not surpass 100%.

- Sea ice volume in each category is adjusted proportionally to changes in SIC while preserving the ice thickness.

This approach ensures physical constraint and model stability after the error correction.

## 2.4 Offline error correction approach

The offline error correction approach refers to performing post-processing of the dynamical model predictions (blue line in Figure 1). The ML configuration is the same as the online configuration (section 2.5). The input features are monthly SST, SSS, total SIC, and latitude. The output feature is the error in the monthly SIC. The predicted error is subtracted from the monthly SIC. If the updated monthly SIC falls below 0% or exceeds 100%, it is set to 0% or 100%, respectively. For more details about the offline error correction approach, please refer to Table 1.

It's worth noting that the offline error correction approach targets directly monthly average model outputs, whereas the online error correction approach addresses instantaneous model errors (Figure 1) and indirectly changes the monthly model outputs during the production of predictions. Therefore, their input and output features are different (Table 1).

## 2.5 Machine learning configuration

As mentioned in the previous sections, the ML model configurations employed for online and offline error correction approaches share an identical architecture (i.e., the same number of layers and the same number of neurons in each layer), but differ in the input and output variables, resulting in different numbers of trainable parameters (for more details, please refer to Table 1).

The ML model uses the values from a single grid point as input to predict the value at the same grid point, meaning one ML model for all grid points. This simplifies the training process while still enabling the development of efficient models.

**Table 2.** Number of parameters of the online and offline ML-based error correction models for each ML model.

| | Online ML-based SIC model | Online ML-based SST/SSS model | Offline ML-based SIC model |
|---|---|---|---|
| BatchNorm | 52 | 52 | 20 |
| Dense layer 1 | 840 | 840 | 360 |
| Dense layer 2 | 1830 | 1830 | 1830 |
| Gate layer | 31 | 31 | 31 |
| Dense layer 3 | 840 | 840 | 360 |
| Dense layer 4 | 1830 | 1830 | 1830 |
| Output | 155 | 31 | 31 |

The ML architecture used in this study is a multilayer perceptron (MLP), a fully connected neural network well-suited for capturing complex nonlinear relationships in data. MLP offers several advantages, including flexibility in handling diverse input features, efficient training via backpropagation, and strong generalization when properly regularized. Additionally, MLP is computationally more efficient than complex deep learning architectures such as convolutional neural networks (CNNs) and U-Net. It has been successfully applied to error correction in geophysical modeling (e.g., Yang et al., 2023), as it is computationally efficient and requires less training data (Jia et al., 2019; Watson, 2019).

The entire MLP architecture consists of seven layers:

- **Input layer**: A batch normalization layer (Ioffe, 2017), which helps stabilize and accelerate the training process by normalizing the input features.

- **Second layer**: A dense layer with 60 neurons, using the rectified linear unit (ReLU) activation function.

- **Third layer**: A dense layer with 30 neurons, also employing the ReLU activation function. This layer shares the same structure as the second layer.

- **Fourth layer**: An attention mechanism implemented via a gate layer, which enables the model to focus on important features, thereby enhancing learning efficiency and predictive performance.

- **Fifth layer**: A dense layer with 60 neurons and ReLU activation, mirroring the configuration of the second layer.

- **Sixth layer**: A dense layer with 30 neurons and ReLU activation, identical to the third layer.

- **Output layer**: A dense layer activated by the linear function.

The objective function used in this study is the mean squared error (MSE). Additionally, details regarding the number of parameters for each ML model are provided in Table 2. To reduce the risk of overfitting and improve model generalization, the following strategies are implemented:

- **Batch Normalization**: The inputs of each layer are normalized to reduce internal covariate shift, thus promoting training stability and generalization.

- **L2 Regularization**: A penalty is applied to the output layer weights, effectively discouraging over-complex models and reducing the likelihood of overfitting.

- **Early Stopping**: The validation loss is monitored during training and the training is halted once the validation loss curve does not decline, avoiding overfitting due to the training data.

To achieve better training results, we further implement the following settings:

- We adopt a running training strategy, using data from the 11 years preceding the test set to train the ML models. For instance, to develop error correction models for predictions in 2011 (a test set), we train the model using data from 2000 to 2009 and validate it with data from 2010. Similarly, for predictions in 2021, we use data from 2010 to 2019 for training and data from 2020 for validation. This approach ensures that the ML models leverage the most recent data while maintaining a clear separation between training, validation, and test sets. The primary reason for using running training is the pronounced decline trend in Arctic sea ice observed over recent decades, with substantial differences between earlier ice conditions (e.g., the 1980s) and those of recent years (e.g., the 2010s). We also performed sensitivity studies on the length of the running training set (e.g., the most recent 5 years or all years since 1980) and the comparison between the running training and the fixed-period training (1992-2002), which are not shown in the paper. We found that the data from the most recent 11 years leads to the best performance for ML training, and the running training outperforms the fixed-period training.

- The characteristic of model errors varies with the calendar month. For instance, the model errors mainly appear in the marginal zone in winter but in the entire sea ice-covered region in summer. We train separately for each calendar month, leading to a distinct ML model for each calendar month. This results in 236 neural network models (from February 2003 to September 2022 based on test months) for the online case. In the offline case, we also consider the start month, resulting in 836 (4 initialized months × 11 lead months × 19 test years) models. Despite the large number of models, the training process is highly efficient due to the simple architecture and low data dimensionality. As a result, training each model is very quick, taking only one minute on a CPU, making this exhaustive approach computationally affordable.

- We train and apply error correction models to grid points where the total SIC exceeds $1\%$. It avoids adding sea ice into open water areas and thus dynamical inconsistency. It also means that our correction models can not create ice on a grid point where the model predicted ice-free conditions.

## 2.6 Hindcast experiments

The standard hindcasts (hereafter referred to as **Reference**) are initialized from the reanalysis presented in section 2.2 in the middle of January, April, July, and October each year, spanning from 1992 to 2021, with a duration of 12 months. From 1992

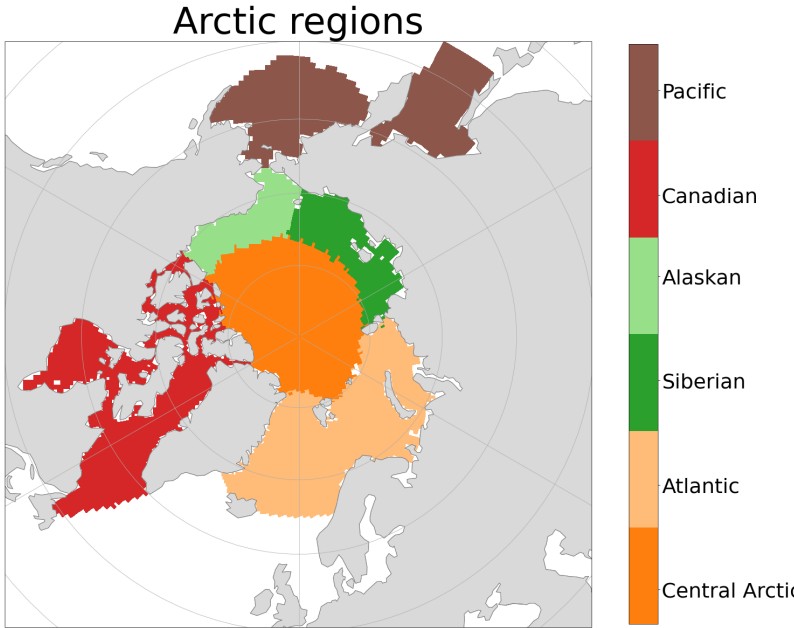

**Figure 2.** Regional domain definitions for central Arctic, Atlantic, Siberian, Alaskan, Canadian, and Pacific regions are based on sea area definitions in Kimmritz et al. (2019) and are similar to those used in Bushuk et al. (2024). Atlantic region: Greenland, Ice, Norwegian, Barents and Kara Seas; Siberian region: Laptev and East Siberian Seas; Alaskan region: Chukchi and Beaufort Seas; Canadian region: Canadian archipelago, Hudson Bay, Baffin Bay, and Labrador Sea; Pacific region: Bering Sea and the Sea of Okhotsk.

to 2002, the first 9 ensemble members of the 30-member reanalysis are used to carry out the hindcast experiments, while after
235 2003, the first 10 ensemble members are used to initialize the hindcast experiments. It is worth noting that these differences (i.e., the different ensemble sizes) would have minimal impact on the results of this study.

A new set of hindcasts (hereafter referred to as **OnlineML**), similar to Reference but with the online error correction approach (section 2.3), are initialized from the reanalysis in the middle of January, April, July, and October from 2003 to 2021. In the production of each hindcast, NorCPM pauses in the middle of each lead month and uses the online error correction model
to predict the error correction and then update the instantaneous model state.

The offline error correction approach (section 2.4) is applied to post-process the hindcasts of Reference (hereafter referred to as **OfflineML**).

## 2.7 Metrics for evaluation

SIE is a commonly used metric in seasonal sea ice prediction (e.g., Bushuk et al., 2024). We evaluate the prediction skill of SIE
in the Pan-Arctic and six Arctic regions depicted in Figure 2. These regional definitions adhere to the area definitions provided by Kimmritz et al. (2019), albeit with the consolidation of the original fourteen sea areas into six regions that are very similar to the ones used in Bushuk et al. (2024). In this study, the SIE is defined as the total area of all grid points within the region

of interest where SIC $\geq 15\%$. SIE is calculated for each ensemble member, and we evaluate the ensemble mean by averaging SIE across all ensemble members.

To evaluate the performance of the ML-based error prediction models, we employ the mean absolute error (MAE), defined as:

$$\text{MAE} = \frac{1}{M} \sum_{i=1}^{M} |\mathbf{E}_\text{p} - \mathbf{E}_\text{t}|, \tag{4}$$

where $\mathbf{E}_\text{p}$ denotes the predicted error and $\mathbf{E}_\text{t}$ denotes the "true" error. In more detail, $\mathbf{E}$ refers to the SIC error at each grid point over the entire evaluation period. $M$ represents the total number of data points used in the MAE calculation.

To evaluate the sea ice prediction skill, we employ the root mean square error (RMSE) as follows:

$$\text{RMSE} = \sqrt{\frac{1}{N} \sum_{i=1}^{N} (\mathbf{X}_\text{p} - \mathbf{X}_\text{t})^2}, \tag{5}$$

where $\mathbf{X}_\text{p}$ represents the prediction and $\mathbf{X}_\text{t}$ represents the "truth" (i.e., the reanalysis in this study). In this study, $\mathbf{X}$ can refer to either the integrated ice-edge error (IIEE) on a Pan-Arctic scale, the SIE on a Pan-Arctic/regional scale, or the SIC at a specific grid point. $N$ represents the number of hindcasts, spanning from 2003 to 2021.

The IIEE is also a crucial metric for sea ice predictions (Goessling et al., 2016). It specifically captures the discrepancies along the ice edge by quantifying the area where the predicted and "true" SIC differ significantly. This makes IIEE particularly valuable for evaluating the spatial accuracy of the ice edge location, offering insight into the performance of models in reproducing the dynamic boundary between ice-covered and open ocean regions. Following the definition of Goessling et al. (2016), the IIEE is computed as the area where the prediction and the "truth" disagree on the SIC being above or below 15%:

$$\text{IIEE} = \int_A \max(c_\text{p} - c_\text{t}, 0) \, dA + \int_A \max(c_\text{t} - c_\text{p}, 0) \, dA, \tag{6}$$

where $A$ is the area of grid cell, $c = 1$ where SIC is above $15\%$ and $c = 0$ elsewhere, and subscripts $p$ and $t$ denote the prediction and the "truth". The definition of the IIEE is equivalent to the so-called symmetric difference between the areas enclosed by the predicted and "true" ice edges.

     To evaluate the significance of prediction skill difference, we use a two-tailed Student's t-test to compare the IIEE or the RMSE between two predictions.

To estimate the uncertainties in an RMSE value arising from the small ensemble size, we employ the bootstrap method. Specifically, we randomly sample 10 ensemble members with replacement from the ensemble, compute the ensemble mean, and then calculate the RMSE (for either SIC or SIE) based on this resampled data. This process is repeated 10,000 times, producing a distribution of 10,000 RMSE values. The standard deviation of this distribution is then used to quantify the 275 uncertainties associated with the RMSE value.

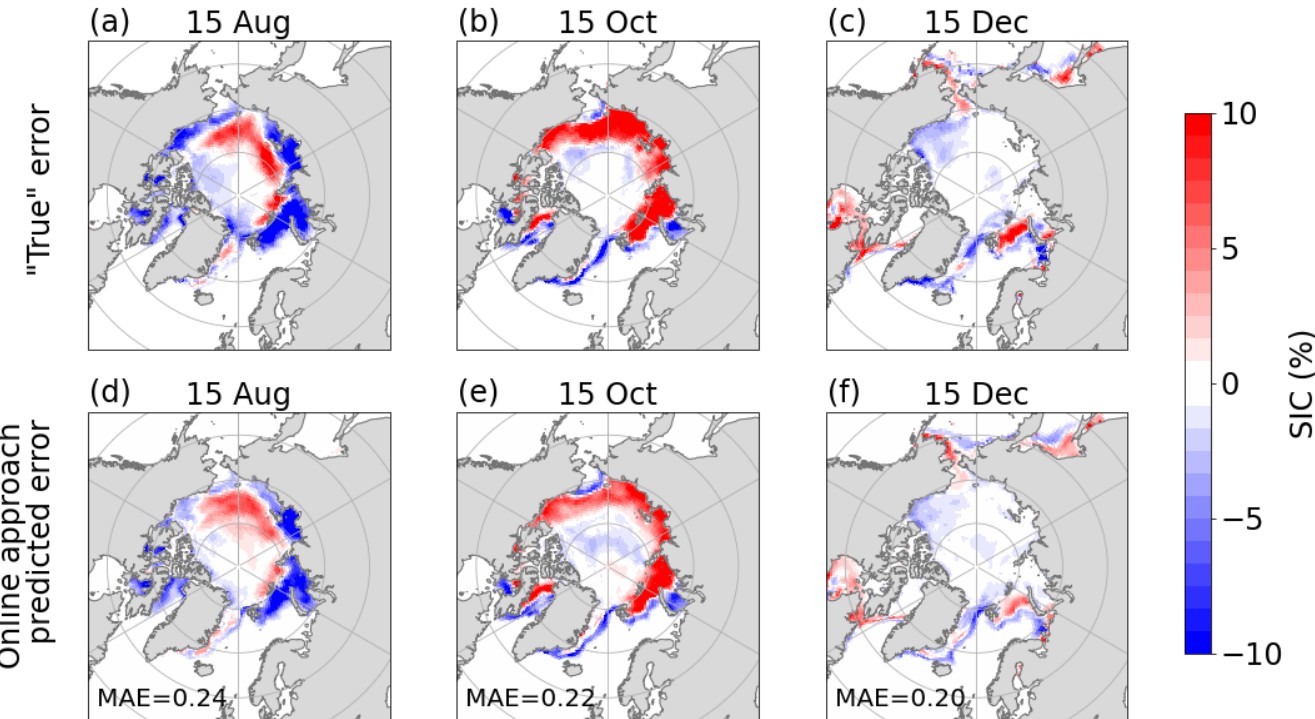

**Figure 3.** Top row: "true" errors of SIC in the middle of the month based on the analysis increments (i.e., the changes thanks to monthly DA in the reanalysis). Bottom row: the errors predicted by the online error correction model. These errors are averaged over the period 2003-2021. Values in the bottom row are the MAE between the "true" and predicted errors across space.

## 3 Results

### 3.1 Error correction model performance

We first demonstrate the performance of ML-based error correction models in predicting the model errors.

The "true" errors obtained from analysis increments and the errors predicted by the online error correction model are av-
eraged over 2003-2021 and displayed in Figure 3. The spatial patterns of the "true" error vary significantly across different dates. For instance, on August 15, errors are predominantly negative across most regions as NorCPM underestimates SIC in this month, with some localized positive errors occurring internally. The average MAE across ice-covered grid points is $0.24\%$. On October 15, the errors are mostly positive as NorCPM overestimates SIC, resulting in an average MAE of $0.22\%$. On December 15, the MAE is $0.20\%$, primarily appearing in marginal ice areas, with overall lower magnitudes compared to August
and October. Notably, the average error remains below $1\%$ in all cases.

For all those months and regions, the online error correction models can correctly predict the spatial pattern of the "true" error (Figure 3). Also, the magnitude of the "true" error is well reproduced with a slight underestimation.

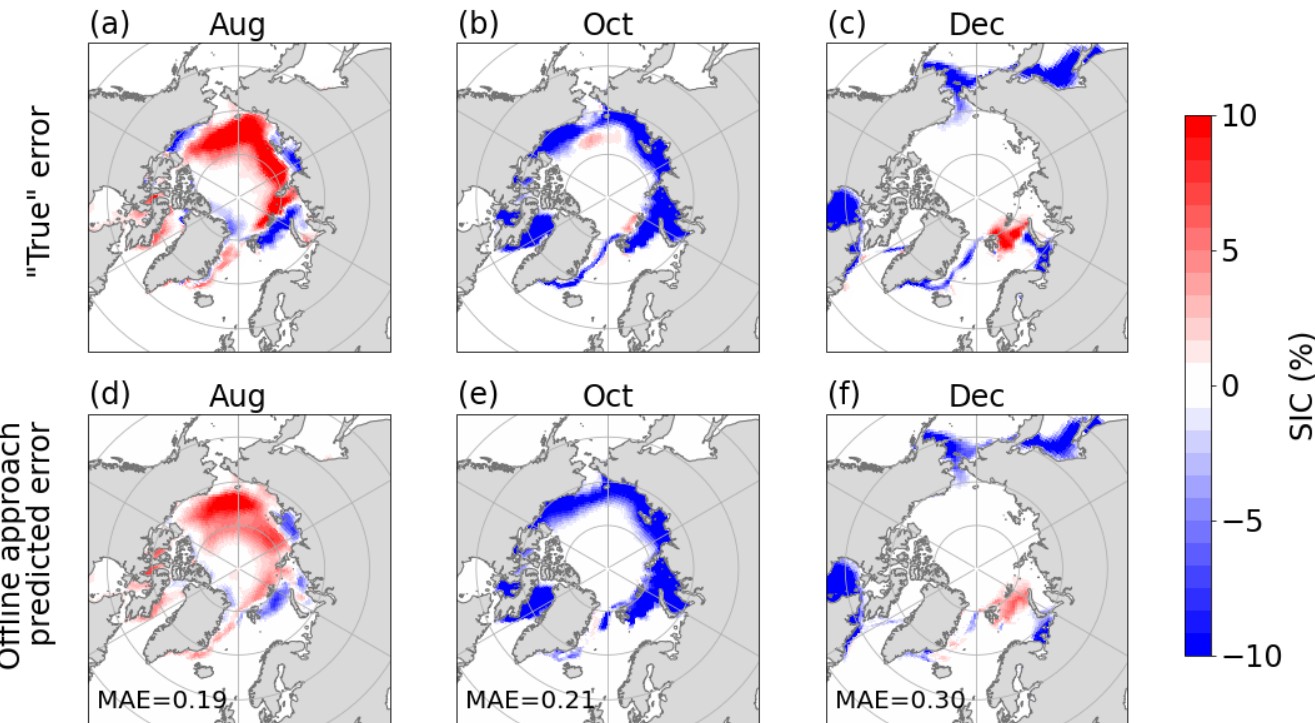

**Figure 4.** Top row: "true" errors of monthly SIC estimated by the Reference hindcast initialized in July minus the reanalysis. Bottom row: the errors predicted by the offline approach. The errors are averaged over the period 2003-2021. Values in the bottom row are the MAE between the "true" and predicted errors across space.

To assess the offline error correction model, we show its performance for hindcasts initialized in July (Figure 4). The monthly "true" error patterns vary significantly across months. The offline error correction models effectively predict the spatial pattern of the "true" errors (Figure 4). The predicted error magnitude is similar to that of the "true" error, with only a slight underestimation. Notably, the MAE of the offline error correction approach is higher than that of the online error correction approach in December (0.30% versus 0.20%), which can be attributed to the model divergence since the initialization in July.

In summary, the above results suggest that the ML-based error correction models in both online and offline scenarios can skillfully predict the large-scale spatial patterns of the SIC error, but slightly underestimate its magnitude.

## 3.2 Application to seasonal predictions

### 3.2.1 Skill seasonality

This section assesses the three sets of hindcasts initialized in January, April, July, and October from 2003 to 2021. The ensemble hindcasts are initialized with the first 10 members of the reanalyses and predict for 11 months (section 2.6).

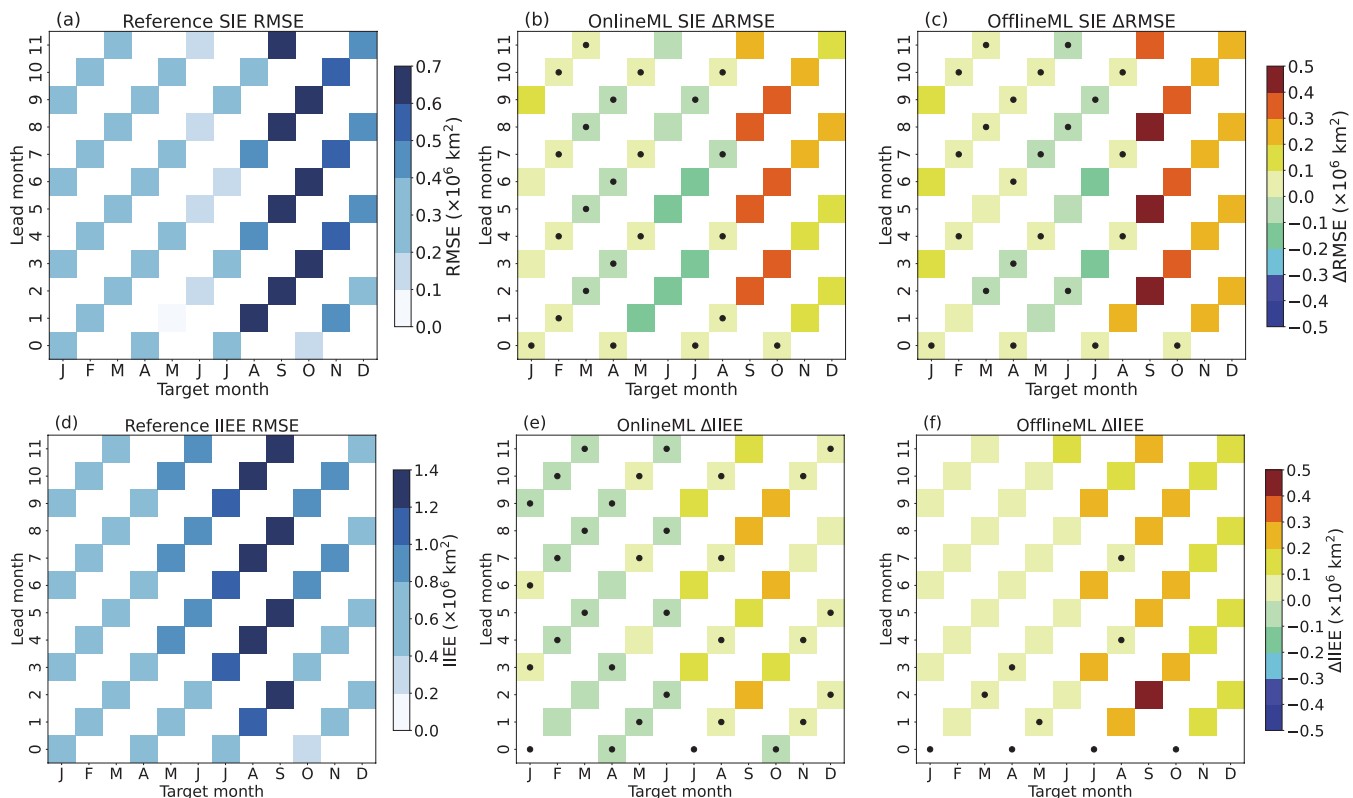

**Figure 5.** (a) RMSE of SIE for the Reference hindcast, (b) ΔRMSE between the Reference and OnlineML hindcasts, (c) ΔRMSE between the Reference and OfflineML hindcasts. (d) IIEE of the Reference hindcast, (e) ΔIIEE between the Reference and OnlineML hindcasts, (f) ΔIIEE between the Reference and OfflineML hindcasts. In b,c,e, and f, warm colors (red/yellow) indicate that the OnlineML or OfflineML hindcasts are better than the Reference hindcasts, while cold colors (blue/green) indicate they are worse than the Reference hindcast. The black dots represent regions where the ΔRMSE or ΔIIEE does not pass the 95% significance test.

Figure 5 presents a comparative analysis of the RMSE for SIE prediction and the IIEE for ice edge prediction in the Pan-Arctic across the three hindcast sets. The Reference hindcast shows higher RMSE in September and October (Figure 5a), primarily due to several factors that have been documented in Bentsen et al. (2013). NorCPM overestimates the Arctic cloudiness, and its summer-season snowmelt is too slow. In addition, NorCPM has slightly too weak winds across the polar basin. These factors lead to too thick sea ice in the polar oceans and excessive Arctic SIE, in particular in summer (Bentsen et al., 2013).

Both the OnlineML and OfflineML hindcasts exhibit a small error reduction from January to July and a large error reduction from August to December (Figure 5b and 5c). The OnlineML hindcast, in which only SIC, SST, and SSS are corrected without directly adjusting the atmospheric component, shows some improvements, particularly in January and from September to December. In contrast, from February to August, the Reference hindcast already exhibits good performance, leading to no significant differences. Compared to the OnlineML hindcast, the OfflineML hindcast achieves a greater error reduction,

particularly in September, where they reduce the SIE prediction error by up to $75\%$ relative to the Reference hindcast. The primary reason is that the online approach corrects instantaneous model errors (on the 15th day of the month). Still, during the one-month model integration, the sea ice component dynamically interacts with the other components, leading to error growth. In terms of monthly averaged model outputs, the correction magnitude is damped. In contrast, the offline approach aims to directly post-process monthly outputs without model integration.

The IIEE shows similar results to the RMSE of SIE (Figure 5d-f). For the Reference hindcast, the IIEE is higher from July to September. The online approach leads to some improvements over the Reference hindcast from July to December, but its error reduction is small or not significant in the other months. In contrast, the offline approach consistently improves performance across nearly all periods and achieves larger error reductions in IIEE than the online approach, particularly from June to January, with the maximum reduction exceeding $0.5 \times 10^6$ $km^2$ compared to the Reference hindcast. By directly correcting monthly

mean outputs, the offline approach avoids information loss during the model integration, leading to larger error reduction.

In summary, the Reference hindcast exhibits relatively larger prediction errors from August to October, primarily due to increased model uncertainties associated with atmospheric forcing and sea ice processes. The offline approach outperforms the online approach in reducing both the RMSE of SIE and the IIEE along the ice edge, particularly during high-error months. For example, in September, the RMSE of SIE is reduced by $75\%$, and the IIEE is reduced by over $0.5 \times 10^6$ $km^2$ compared to the

325 Reference hindcast.

### 3.2.2 Skill of seasonal predictions for different regions

The previous section highlighted significant improvements in predictions, primarily evident from September to January regardless of the initialized month. In this section, we focus on analyzing the hindcasts initialized in July, and we show the performance for different regions and both SIE and SIC. It is mostly because summer sea ice prediction serves as a critical

climate change indicator, affects ecosystems and human activities, and presents a significant scientific challenge due to its high variability (Figure 5a). For validation on the other initialization months, please refer to Figures S1-S4.

We first investigate the seasonal prediction skill for Pan-Arctic and regional SIE defined in Figure 2. For the Pan-Arctic SIE, previously assessed in Figure 5, both the OnlineML and OfflineML hindcasts reduce the SIE RMSEs (Figure 6a). The RMSEs in the OnlineML hindcast have a strong seasonality as that in the Reference hindcast: higher in August, September,

and October, and lower in November, December, and January. The OfflineML hindcast has the lowest RMSEs, in particular, an RMSE reduction of about 75% compared to the Reference hindcast in September.

Both error correction approaches reduce the RMSEs for regional SIE, and the offline approach overall outperforms the online approach (Figure 6b-f). In the Atlantic region (Figure 6b), significant RMSE reduction is observed for the first three months, until October. The OfflineML hindcast has the lowest RMSEs until September and similar RMSEs to the OnlineML

hindcast from October. In the Siberian region (Figure 6c), the RMSE reduction due to error correction is significant only until October but becomes almost zero from November due to the region being fully covered by sea ice. The OfflineML hindcast is significantly better than the OnlineML hindcast until September and similar afterward. In the Alaskan region (Figure 6d), there is no significant RMSE reduction in August, but we observe significant RMSE reductions from September to November.

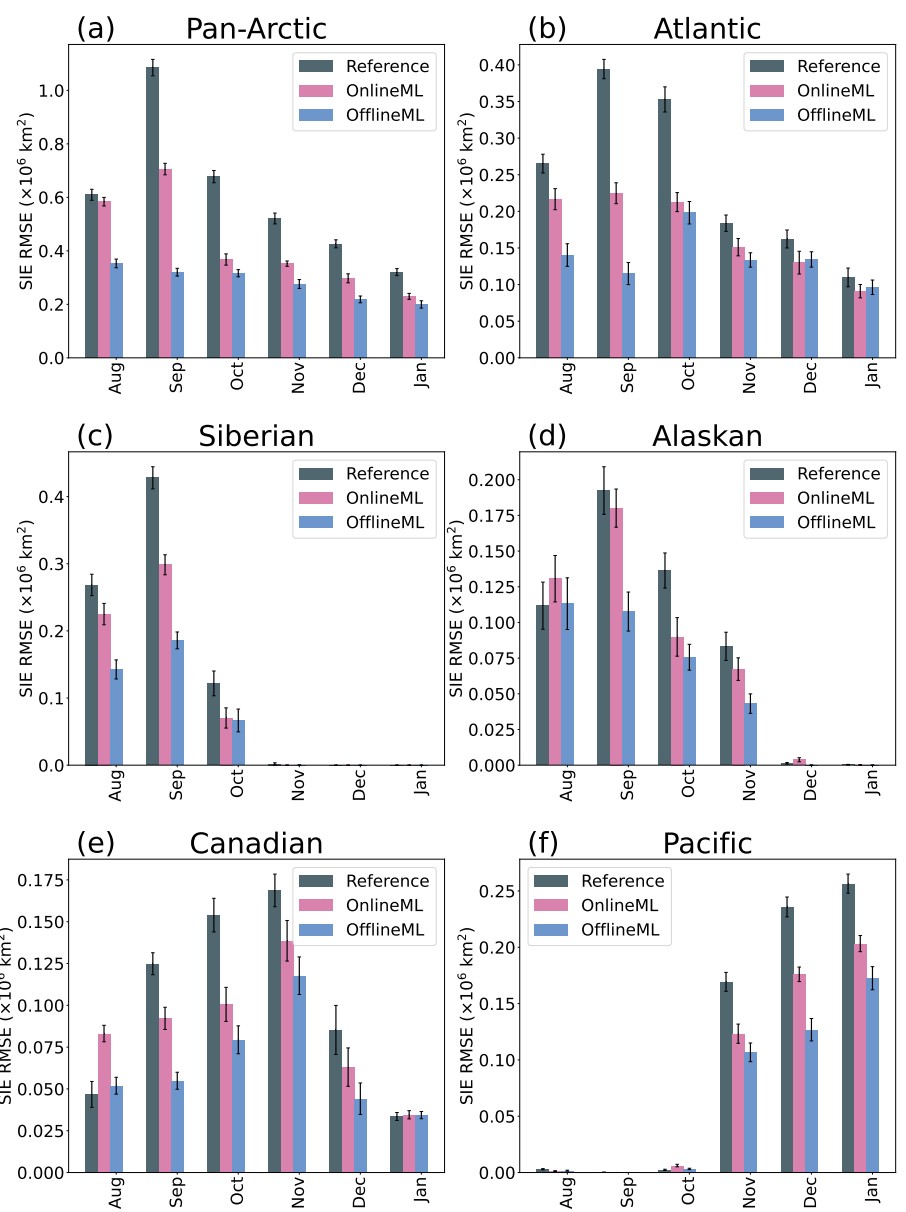

**Figure 6.** RMSE of SIE in the Pan-Arctic and five subregions for the Reference hindcast (gray bar), the OfflineML hindcast (blue bar), and the OnlineML hindcast (purple bar). Error bars represent the uncertainties.

In December and January, the region is almost fully covered by sea ice, leading to very tiny RMSEs for all three hindcast experiments. In the Canadian region (Figure 6e), both approaches lead to significant RMSE reductions from September to December and the offline approach outperforms the online approach. In addition, the online approach leads to a significantly larger RMSE in August than that of the Reference hindcast. In the Pacific region, the RMSEs are near zeros from August to October due to very limited sea ice coverage. The two error correction approaches lead to significant RMSE reductions after November, and the offline approach outperforms the online approach in December and January.

Notably, in August, the RMSE of the OnlineML hindcast exceeds that of the Reference hindcast in both the Alaskan and Canadian regions. This is primarily due to the systematic underestimation of SIE by the OnlineML hindcast relative to both the Reference hindcast and the "truth" in these regions (Figure S5). The underlying causes of this systematic underestimation, however, warrant further investigation.

The offline approach outperforms the online approach across all regions, primarily because the online correction targets instantaneous model errors (i.e., those on the 15th day of each month). These corrected errors may reemerge through interactions with the other components of the coupled model system, thereby diminishing the overall impact of the online error correction when evaluated using monthly-averaged outputs. In contrast, the offline approach directly adjusts the monthly model outputs, which aligns closely with the evaluation metrics used in this study. Moreover, the offline approach does not need to run the dynamical model and is computationally cheaper than the online approach. However, the online approach not only reduces SIC errors but also propagates corrections through the model integration to the other variables (e.g., sea ice thickness and sea ice drift), ensuring physical consistency between the predicted variables.

In summary, while the error correction performance varies by region and target month, overall, both approaches improve the sea ice prediction. In addition, the offline approach is more efficient than the online approach in reducing the SIE RMSE for both Pan-Arctic and subregions. These conclusions also hold for seasonal predictions initialized in the other seasons. For details, please refer to Figures S1-S4.

We take a closer look at the spatial aspects of the offline error correction approach in hindcasts initialized in July (Figure 7). We specifically focus on identifying local areas where the error correction leads to improvements that may not be evident when examining SIE alone.

The impact of the error correction on SIC is more pronounced near the ice edge (Figure 7). In August, only a few grid points in the Siberian and Atlantic regions exhibit improvements (Figure 7a), with an average improvement of 0.99%. In September, notable enhancements appear across the central Arctic, Atlantic, Siberian, and Canadian regions (Figure 7b), with an average improvement of 6.52%. In October, significant improvements are observed in the Atlantic and Canadian regions, reaching an average of 10.22%. Additionally, some blue areas emerge in the central Arctic, indicating substantial differences between the Reference hindcast and the OfflineML hindcast, though the magnitude of these differences remains minimal. In November and December, the positive impact of the error correction is primarily concentrated in Hudson Bay and the Sea of Okhotsk. However, noise increases in the central Arctic, and the average improvement declines to 2.13% in November and 1.71% in December. The widespread presence of blue in the central Arctic also results in an average improvement of $-0.08\%$ in January.

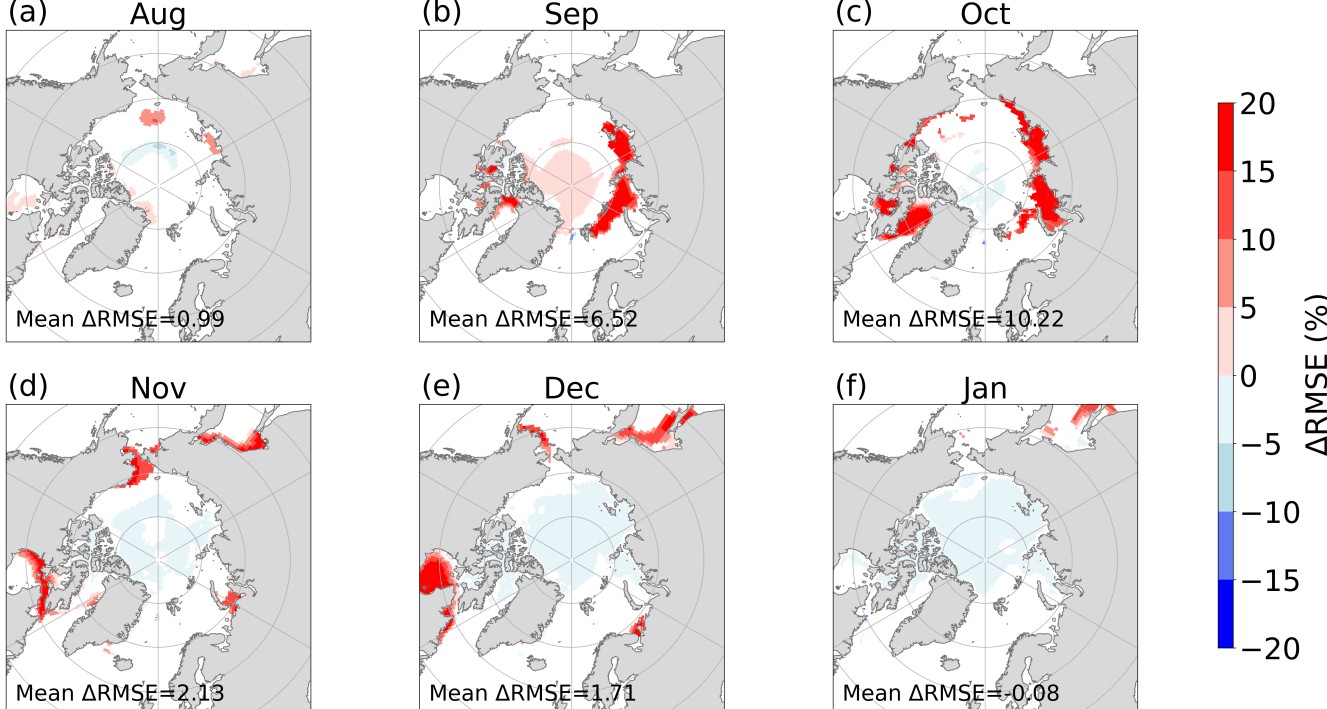

**Figure 7.** Differences in SIC RMSE between the Reference and OfflineML hindcasts initialized from July. Warmer (colder) colors indicate that the OfflineML hindcast outperforms (underperforms) the Reference hindcast. White areas indicate differences that are not statistically significant. The value in each subplot represents the average ΔRMSE for statistically significant grid points.

The major improvements in SIC are evident near the ice edge, which is closely associated with SIE. This spatial distribution highlights how the ML-based error correction approaches can enhance model performance in different regions, particularly during the ice-advance season, where using SIE as a metric might obscure these localized gains. In addition, it is noteworthy that the OfflineML and OnlineML hindcasts exhibit similar error spatial distributions. For specific details about the OnlineML hindcast, please refer to Figure S6.

## 4 Discussions and conclusions

In this study, we apply ML within NorCPM, a fully coupled Earth system model, to improve seasonal sea ice predictions in the Arctic under both online and offline scenarios. In the online error correction approach, ML is utilized to rectify errors in instantaneous model states in the middle of the month that serve as initial conditions for the subsequent model integration. The offline error correction approach involves the post-processing of monthly sea ice predictions.

The approaches proposed in this study integrate ML with a dynamical modeling framework, with the primary objective of reducing the intrinsic prediction errors of the dynamical model itself. Unlike purely data-driven models (e.g., Andersson et al.,

2021; Ren et al., 2024; Kim et al., 2025), which are typically designed for statistical prediction of specific sea ice properties, ML here aims to improve the overall performance of the dynamical prediction system that ensures physical consistency among a large number of predicted variables.

Our results demonstrate that both online and offline ML-based error correction models can well predict the spatial distribution of errors, albeit with slight deficiencies in capturing amplitude. By applying the two approaches to seasonal Arctic sea ice predictions initialized from January, April, July, and October, we find that both approaches can reduce SIE and IIEE prediction errors compared to the raw predictions without error correction. The improvements vary with the lead month, with particularly notable enhancements observed in predictions from August to October.

By comparing the two error correction approaches, we find that the offline approach yields smaller errors than the online approach. The online error correction approach corrects instantaneous model errors only on the 15th day of the month, and the effect of this correction gradually weakens during model integration due to the accumulation of errors in the other model components. Consequently, the impact of the correction becomes less evident when computing monthly-averaged outputs. Nevertheless, the online error correction can reduce errors in SST and SSS (Figures S7 and S8). Moreover, the online correction approach maintains better physical consistency among the predicted variables through dynamical model integration. The offline error correction approach directly corrects the model outputs without requiring model integration. As a result, it is computationally more efficient and easier to integrate into operational sea ice prediction systems than the online approach.

It is important to note that the proposed approaches still have room for improvement. In this study, we only use ocean and sea ice variables as input features. Including atmospheric variables would help to address errors due to both dynamic and thermodynamic processes and further improve the performance. Increasing the frequency of online correction could help enhance its effectiveness (Gregory et al., 2024), but this is challenging in practice since analysis increments in NorCPM are currently available only every month. An alternative strategy is to train hybrid models that combine ML with dynamical models, which has been shown to be effective in other systems (Farchi et al., 2021). However, this approach relies on external constraints to compute the gradient of the dynamical model, which are not available in NorCPM.

Furthermore, the current ML model (MLP) is trained independently at each grid point and thus cannot capture spatial correlations. This limits its ability to correct spatially coherent errors, particularly in regions where NorCPM already performs well and only subtle adjustments are needed. As a result, the hybrid model often struggles to reproduce the reanalysis, which are treated as the "truth" in this study. While it is unrealistic to expect the model to perfectly replicate analysis increments, the discrepancy is closely related to the ML-based model's learning capacity and the nature of the underlying errors. Possible contributing factors include: (1) the lack of spatial dependencies due to pointwise training and (2) the tendency of models trained on long-term data to learn systematic biases rather than instantaneous random errors, the latter of which tend to be averaged out over time. Therefore, there is still room to improve the ML-based error correction framework. Future studies could explore spatially-aware architectures, such as CNNs and U-Net, and incorporate additional predictors to capture complex error structures and enhance correction performance (Palerme et al., 2024).

*Code and data availability.* The code and data to plot figure is available at https://doi.org/10.5281/zenodo.14533027.

*Author contributions.* Conceptualization: ZH, YW, JB. Analysis and Visualization: ZH. Interpretation of results: ZH, YW, JB. Writing
(original draft): ZH, YW. Writing (reviewing and editing original draft): ZH, YW, JB, XW, ZS.

*Competing interests.* The author declares that no competing interests.

*Acknowledgements.* This study was funded by National Natural Science Foundation of China (42225602), the National Key $R\&D$ Program of China (2022YFE0106400), Postgraduate Research & Practice Innovation Program of Jiangsu Province (KYCX23_0657), the China Scholarship Council (202206710071), the Special Founds for Creative Research (2022C61540), the Opening Project of the Key Laboratory
of Marine Environmental Information Technology (521037412). JB was funded by the project TARDIS funded by the Research Council of Norway (under grant No. 325241). YW was funded by the Norges Forskningsråd (under grant No. 328886) and the Trond Mohn stiftelse (under grand No. BFS2018TMT01). This work also received grants for computer time from the Norwegian Program for supercomputer (NN9039K) and storage grants (NS9039K). ZS was funded by the National Natural Science Foundation of China (42176003).

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
