# Peer review of "Correcting Errors in Seasonal Arctic Sea Ice Prediction of Earth"

_EGUsphere, 2024_

## Author Comment (AC1)

**Improving Seasonal Arctic Sea Ice Predictions with the Combination of Machine Learning and Earth System Model**

Addressed Comments for Publication to

**The cryosphere**

by

Zikang He, Yiguo Wang, Julien Brajard, Xidong Wang, Zheqi Shen

**Authors' Response to Reviewer #1**
* * *
**Comment 1**

The authors have improved the seasonal prediction skill of Arctic sea ice in the NorCPM model using a machine learning method. Specifically, they experimented with two approaches: (a) an online method, which involves correcting the model's initial fields after each time step, and (b) an offline method, where the model runs freely, and the results are uniformly corrected afterward. Both methods are based on simple concepts but have been proven to be effective.

I acknowledge that the approach of integrating machine learning with dynamical models is quite advanced and interesting; however, I have some concerns regarding the broader scientific implications of this study. Please see the following general comments.
* * *
**Response:** We very much appreciate that the reviewer found this study advanced and interesting. We thank the reviewer for providing insightful comments that helped significantly improve the manuscript. We carefully addressed each concern and revised the manuscript. Below, we provide our detailed point-by-point responses to the reviewer's comments. To enhance the legibility of this response letter, the reviewer's comments are typeset in blue boxes. Rephrased or added statements in the revised version of the manuscript are indicated in gray boxes.

**General Comments**
* * *
**Comment 2**

The discussion section is currently somewhat underdeveloped. I would suggest that the authors discuss their results from the following perspectives:

a) How do the findings of this study inform physical enhancements to predictive systems? For instance, which sea ice processes benefit from online correction of sea ice concentration? And thus improved prediction skills?
* * *
**Response:** We thank the reviewer for the suggestion. In this study, all sea ice errors associated with thermodynamic and dynamic processes were incorporated into the error correction process. Still, we did not explicitly attribute sea ice errors to specific physical processes. Specifically, the online correction method simultaneously adjusts sea ice concentration (SIC), sea surface temperature (SST), and sea surface salinity (SSS). While updating SST primarily addresses thermodynamic errors, changing SST and SSS would change the water density and influence the ocean circulation.

For clarity, we have revised one paragraph in the discussion and conclusion section as follows (L406-L412 in the manuscript):
* * *
It is important to note that the proposed approaches still have room for improvement. In this study, we only use ocean and sea ice variables as input features. Including atmospheric variables would help to address errors due to both dynamic and thermodynamic processes and further improve the performance. Increasing the frequency of online correction could help enhance its effectiveness (Gregory et al., 2024), but this is challenging in practice since analysis increments in NorCPM are currently available only every month. An alternative strategy is to train hybrid models that combine ML with dynamical models, which has been shown to be effective in other systems (Farchi et al., 2021). However, this approach relies on external constraints to compute the gradient of the dynamical model, which are not available in NorCPM.
* * *
> ### Comment 3
>
> b) As a hybrid approach integrating dynamical models with machine learning, what advantages does this study offer compared to purely data-driven machine learning methods (e.g., Andersson et al., 2021; Ren et al., 2024; Kim et al., 2025)? For instance, does it demonstrate enhanced scalability, such as in applications to ice thickness prediction corrections?

**Response:** The primary objective of the approaches used in this study is to reduce the intrinsic prediction errors of the dynamical model itself. The dynamical model can provide predictions for a large number of variables and ensures physical consistency among the variables. On the other hand, purely data-driven models, typically relying on predictors from observations or reanalysis data, aim for certain specific variables and often lack physical consistency among these variables.

As suggested, we have added one paragraph to the discussions and conclusions section as follows (L388-L392 in the manuscript):

> The approaches proposed in this study integrate ML with a dynamical modeling framework, with the primary objective of reducing the intrinsic prediction errors of the dynamical model itself. Unlike purely data-driven models (e.g., Andersson et al., 2021; Ren et al., 2024; Kim et al., 2025), which are typically designed for statistical prediction of specific sea ice properties, ML here aims to improve the overall performance of the dynamical prediction system that ensures physical consistency among a large number of predicted variables.

> ### Comment 4
>
> c) The current training framework employs a relatively short rolling forecast window (10-month input $\longrightarrow$ 1-month output). However, would a simple MLP's nonlinear approximation capacity remain effective when applied to longer windows required for daily sub-seasonal timescale predictions (e.g., 90-day input $\longrightarrow$ 1-day output)? Alternatively, might this would require more comprehensive deep learning methods? I recommend expanding the discussion to explicitly address the method's generalizability across varying temporal scales.

**Response:** We thank the reviewer very much for this comment. As stated in the manuscript, we used the latest 11 years of data for training and validation for each month (1-month input $\longrightarrow$ 1-month output, L153-L155 in the manuscript):

> Considering the seasonality of the error of the sea ice state, we build one error correction model for each calendar month. Also, we employ a running training strategy and use the most recent 11 years of data before the prediction month (the first 10 years for training and the last year for validation).

As suggested, we have revised the relevant paragraph in the discussions and conclusions section (L406-L422 in the manuscript):

> It is important to note that the proposed approaches still have room for improvement. In this study, we only use ocean and sea ice variables as input features. Including atmospheric variables would help to address errors due to both dynamic and thermodynamic processes and further improve the performance. Increasing the frequency of online correction could help enhance its effectiveness (Gregory et al., 2024), but this is challenging in practice since analysis increments in NorCPM are currently available only every month. An

alternative strategy is to train hybrid models that combine ML with dynamical models, which has been shown to be effective in other systems (Farchi et al., 2021). However, this approach relies on external constraints to compute the gradient of the dynamical model, which are not available in NorCPM.

Furthermore, the current ML model (MLP) is trained independently at each grid point and thus cannot capture spatial correlations. This limits its ability to correct spatially coherent errors, particularly in regions where NorCPM already performs well and only subtle adjustments are needed. As a result, the hybrid model often struggles to reproduce the reanalysis, which are treated as the "truth" in this study. While it is unrealistic to expect the model to perfectly replicate analysis increments, the discrepancy is closely related to the ML-based model's learning capacity and the nature of the underlying errors. Possible contributing factors include: (1) the lack of spatial dependencies due to pointwise training and (2) the tendency of models trained on long-term data to learn systematic biases rather than instantaneous random errors, the latter of which tend to be averaged out over time. Therefore, there is still room to improve the ML-based error correction framework. Future studies could explore spatially-aware architectures, such as CNNs and U-Net, and incorporate additional predictors to capture complex error structures and enhance correction performance (Palerme et al., 2024).

**Comment 5**

d) Comparing online and offline methods based on predictive skill metrics is necessary but may be insufficient. Could the analysis be extended to incorporate additional dimensions to better delineate their applicable scenarios? This expanded discussion would provide more actionable guidance for researchers applying machine learning to calibrate model forecasts in operational settings.

**Response:** We agree with the reviewer on this comment. We have added some discussions on physical consistency and computational cost and have revised the relevant paragraph of the discussions and conclusions section as follows (L398-L412 in the manuscript):

By comparing the two error correction approaches, we find that the offline approach yields smaller errors than the online approach. The online error correction approach corrects instantaneous model errors only on the 15th day of the month, and the effect of this correction gradually weakens during model integration due to the accumulation of errors in the other model components. Consequently, the impact of the correction becomes less evident when computing monthly-averaged outputs. Nevertheless, the online error correction can reduce errors in SST and SSS (Figures S7 and S8). Moreover, the online correction approach maintains better physical consistency among the predicted variables through dynamical model integration. The offline error correction approach directly corrects the model outputs without requiring model integration. As a result, it is computationally more efficient and easier to integrate into operational sea ice prediction systems than the online approach.

It is important to note that the proposed approaches still have room for improvement. In this study, we only use ocean and sea ice variables as input features. Including atmospheric variables would help to address errors due to both dynamic and thermodynamic processes and further improve the performance. Increasing the frequency of online correction could help enhance its effectiveness (Gregory et al., 2024), but this is challenging in practice since analysis increments in NorCPM are currently available only every month. An alternative strategy is to train hybrid models that combine ML with dynamical models, which has been shown to be effective in other systems (Farchi et al., 2021). However, this approach relies on external constraints to compute the gradient of the dynamical model, which are not available in NorCPM.

**Specific Comments**

**Comment 6**

The language needs to be polished, and the logical flow between sentences should be strengthened. The paper's subtitles do not clearly convey the intended meaning and would benefit from revision.

**Response:** We carefully checked the language and logical flow between sentences and believe that the manuscript has been improved. For clarity, we have revised the title of this study as follows:

> Correcting Errors in Seasonal Arctic Sea Ice Prediction of Earth System Model with Machine Learning

**Comment 7**

Lines 9-10: If possible, please use one or two sentences to further elaborate on why the offline error correction approach performs better than the online error correction approach.

**Response:** We thank the reviewer for the comment. As suggested, we have revised the abstract as follows (L10-L13 in the manuscript):

> This is primarily because the online approach targets only instantaneous model errors on the 15th of each month, while errors can grow during the subsequent one-month model integration due to interactions among the model components, damping the error correction in monthly averages.

**Comment 8**

Line 14: Is there a newer paper to cite? This one is not "recent" enough.

**Response:** We have added some other recent references into the manuscript as follows (L16-L17 in the manuscript):

> According to satellite observations, the Arctic sea ice extent (SIE) rapidly declines throughout all calendar months during the recent decades (e.g., Serreze et al., 2007; Onarheim et al., 2018; Wang et al., 2022; Heuzé and Jahn, 2024).

**Comment 9**

Line 19: I suppose the word "compared" could be removed.

**Response:** As suggested, we have removed " and compared" from the text.

**Comment 10**

Line 58: The abbreviation "SIC" appears for the first time without a definition.

**Response:** We thank the reviewer for the comment. We have revised the text as follows (L24-L26 in the manuscript):

They found that dynamical and statistical models are overall comparable in predicting the Pan-Arctic SIE, and dynamical models generally outperform statistical models in predicting the regional SIE and sea ice concentration (SIC, i.e., local quantities).

**Comment 11**

Line 69: The title should be changed to "Data and Methods" as this section introduces the model and data parts first.

**Response:** As suggested, we have revised the section title to "Data and Methods" (L74 in the manuscript).

**Comment 12**

Lines 94: "the summer sea ice extent" should be revised to "the summer SIE".

**Response:** As suggested by the reviewer, we have modified the text as follows (L99-L100 in the manuscript):

Consequently, the summer SIE in the Arctic has large positive biases, contributing to an underestimation of global temperatures (Bentsen et al., 2013; Bethke et al., 2021).

**Comment 13**

Lines 102-103: Maybe I missed something, but why not directly use observations as the "truth"? How much discrepancy is there between the reanalysis of NorCPM and observations?

**Response:** We appreciate the reviewer's comment. The main reason is that NorCPM employs anomaly-field assimilation (Kimmritz et al., 2019; Wang et al., 2019; Bethke et al., 2021), which keeps the model close to its attractor and helps to reduce model drift during the monthly model integration (Carrassi et al., 2014; Weber et al., 2015). The online error correction approach based on the analysis increments of the anomaly-field assimilation is not able to reduce model biases. For clarity, we have revised section 2.1 as follows (L97-L110 in the manuscript):

NorESM1 tends to overly produce thick sea ice, especially in the polar oceans adjacent to the Eurasian continent. This is partly due to factors such as weaker winds across the polar basin and overestimated Arctic cloudiness, which leads to little summer snowmelt. Consequently, the summer SIE in the Arctic has large

positive biases, contributing to an underestimation of global temperatures (Bentsen et al., 2013; Bethke et al., 2021).

NorCPM uses the EnKF to update unobserved ocean and sea ice variables by leveraging state-dependent covariance from the simulation ensemble (Kimmritz et al., 2018, 2019). The EnKF allows the assimilation of observations of various types while accounting for observational errors, spatial coverage, and the evolving covariance with the climate state. The EnKF accounts for uncertainties in initial conditions to generate ensemble predictions, which evolve in time and provide time- and space-dependent error estimates.

NorCPM employs anomaly-field assimilation (Kimmritz et al., 2019; Wang et al., 2019; Bethke et al., 2021) in which the climatology of the observations is replaced by the model climatology calculated from the ensemble mean of the model historical simulation (without assimilation). While the anomaly-field assimilation keeps the model close to its attractor and helps to reduce the model drift during the monthly model integration (Carrassi et al., 2014; Weber et al., 2015), it does not significantly change model biases.

In section 2.2, we have modified the text to provide the reasons for using the reanalysis as the "truth" (L112-L119 in the manuscript):

In this study, we use the reanalysis of NorCPM as the "truth" to assess the improvement achieved by the ML-based error correction approaches. First, it is because NorCPM performs anomaly-field assimilation. The large model biases are not corrected by DA (section 2.1) and thus the analysis increment of the reanalysis used to build the online error correction model (section 2.3) does not take into account model biases. Second, the online error correction approach needs to consistently update SIC in each category, sea surface temperature (SST), and sea surface salinity (SSS) under sea ice, which are often not observed. The reanalysis of NorCPM is a physically consistent construction of the Earth system (Counillon et al., 2016; Kimmritz et al., 2019) and provides a reasonable and physically consistent estimation of these variables. Finally, the reanalysis combining observations with NorESM represents the upper limit of the sea ice predictability of NorCPM.

**Comment 14**

Line 130 & Table 1: Why consider latitude only but not longitude? Can any explanation be provided? And how about the relative importance of these input features?

**Response:** In this study, our error correction approaches take SST, SSS, SIC, and SIT—along with additional latitude information. Longitude is excluded primarily due to technical considerations. Geographically, 0° and 360° represent the same location, but in the ML context, they are treated as two extreme values. To avoid introducing artificial discontinuities or distortions, we chose not to include longitude as an input variable during ML model training.

We apologize to the reviewer for not providing the relative importance of each variable during training, as our ML model training did not track feature importance explicitly. However, Gregory et al. (2023) offered valuable insights into the relative contributions of the predictors, with additional consideration of sea ice velocity components (SIU and SIV). Their findings indicate that SIC alone accounts for approximately 66% of the total prediction skill, making it the most influential input. SST, together with SIU and SIV, contributes an additional 20%, with SST itself estimated to provide around 6–8%. The remaining variables—SIT, shortwave radiation (SW), ice-surface skin temperature (TS), and SSS—collectively account for the final 14%, with SIT and SSS contributing approximately 3–5% and 2–3%, respectively.

**Comment 15**

Lines 142-143: I am curious about the exact post-processing of physically inconsistent fields. Could you give a concise and clear description rather than simply citing a paper?

**Response:** For clarity, we have revised the text as follows (158-L166 in the manuscript):

Before restarting the model after applying online error correction, it is essential to ensure that the updated variables remain within physical limits (e.g., SIC between 0% and 100%) and maintain consistency with non-updated variables. If unphysical values or inconsistencies arise, they can lead to model instability. To prevent these issues, we apply a post-processing method specifically designed for NorCPM (Kimmritz et al., 2018):

- If SIC in any thickness category falls below 0% or exceeds 100%, it is set to 0% or 100%, respectively.

- If the total SIC across all thickness categories exceeds 100%, SIC values in each category are proportionally scaled to ensure the total does not surpass 100%.

- Sea ice volume in each category is adjusted proportionally to changes in SIC while preserving the ice thickness.

This approach ensures physical constraint and model stability after the error correction.

**Comment 16**

Line 205: "Pan-arctic". Please make the capitalization of this term consistent throughout the paper. "Fig. 2" or "Figure 2"? Please also make this consistent throughout the paper.

**Response:** We thank the reviewer for the suggestions. We have gone through the whole manuscript to consistently use "Pan-Arctic" and "Figure X".

**Comment 17**

Lines 217-218: "We define the IIEE as the area where the prediction and the truth disagree on the ice concentration being above or below 15%:" It would be better to rephrase as the IIEE metric has been defined by Goessling et al. (2016). Are the authors themselves defining a new metric called IIEE?

**Response:** We followed the definition of Goessling et al. (2016). For clarity, we have revised the text as follows (L263-L264 in the manuscript):

Following the definition of Goessling et al. (2016), the IIEE is computed as the area where the prediction and the "truth" disagree on the SIC being above or below 15%

**Comment 18**

Lines 211 & 220-221: Why are consistent subscripts not used in these two equations?

**Response:** Sorry for the confusion. To be consistent in subscripts, we have revised the manuscript as follows (L255-L259 in the manuscript):

To evaluate the sea ice prediction skill, we employ the root mean square error (RMSE) as follows:

$$\text{RMSE} = \sqrt{\frac{1}{N}\sum_{i=1}^{N}(\mathbf{X}_\text{p} - \mathbf{X}_\text{t})^2}, \tag{1}$$

where $\mathbf{X}_\text{p}$ represents the prediction and $\mathbf{X}_\text{t}$ represents the "truth" (i.e., the reanalysis in this study). In this study, $\mathbf{X}$ can refer to either the integrated ice-edge error (IIEE) on a Pan-Arctic scale, the SIE on a Pan-Arctic/regional scale, or the SIC at a specific grid point. $N$ represents the number of hindcasts, spanning from 2003 to 2021.

**Comment 19**

Line 223: What is the meaning of "squared errors"? Do you mean "RMSE"?

**Response:** The reviewer is right. We have replaced "squared errors" by "RMSE" in the manuscript (L271 in the manuscript).

**Comment 20**

Line 253: "NorCPM overestimates the Arctic cloudiness, and its summer-season snowmelt is too slow." I am not clear which figure I can draw such a conclusion from.

**Response:** We agree with the reviewer on this comment. The conclusion was drawn from Bentsen et al. (2013). For clarity, we have revised the text as follows (L299-L304 in the manuscript):

Figure 5 presents a comparative analysis of the RMSE for SIE prediction and the IIEE for ice edge prediction in the Pan-Arctic across the three hindcast sets. The Reference hindcast shows higher RMSE in September and October (Figure 5a), primarily due to several factors that have been documented in Bentsen et al. (2013). NorCPM overestimates the Arctic cloudiness, and its summer-season snowmelt is too slow. In addition, NorCPM has slightly too weak winds across the polar basin. These factors lead to too thick sea ice in the polar oceans and excessive Arctic SIE, in particular in summer (Bentsen et al., 2013).

> **Comment 21**
>
> Line 256: "Both the OnlineML and OfflineML hindcasts exhibit similar behaviors regardless of the seasonality." This sentence is somewhat confusing, please rephrase it.

**Response:** For clarity, we have revised the relevant text as follows (L305-L306 in the manuscript):

> Both the OnlineML and OfflineML hindcasts exhibit a small error reduction from January to July and a large error reduction from August to December (Figure 5b and 5c).

> **Comment 22**
>
> Lines 276-277: Why choose to analyze/present the reanalysis initialized in July? Could you provide some explanation? The later analysis should also clarify whether the results shown in Figure 6 depend on the initialization month.

**Response:** For clarity, we have revised the first paragraph of section 3.2.2 as follows (L327-L331 in the manuscript):

> The previous section highlighted significant improvements in predictions, primarily evident from September to January regardless of the initialized month. In this section, we focus on analyzing the hindcasts initialized in July, and we show the performance for different regions and both SIE and SIC. It is mostly because summer sea ice prediction serves as a critical climate change indicator, affects ecosystems and human activities, and presents a significant scientific challenge due to its high variability (Figure 5a). For validation on the other initialization months, please refer to Figures S1-S4.

Also, we have modified the summary paragraph as follows (L362-L365 in the manuscript):

> In summary, while the error correction performance varies by region and target month, overall, both approaches improve the sea ice prediction. In addition, the offline approach is more efficient than the online approach in reducing the SIE RMSE for both Pan-Arctic and subregions. These conclusions also hold for seasonal predictions initialized in the other seasons. For details, please refer to Figures S1-S4.

> **Comment 23**
>
> Lines 289-293 (Figure 6d & 6e): Why is the result of the Online ML hindcast in August worse than the Reference hindcast in the Alaskan and Canadian regions?

**Response:** From the time series of SIE in both regions (Figure R1), the OnlineML consistently underestimates SIE compared to the "truth" and the Reference hindcast. However, the underlying mechanisms responsible for this bias remain unclear at this stage and warrant further investigation.

We also added a description in the manuscript as follows (L350-L353 in the manuscript):

[Figure]

**Figure R1.** Time series of SIE in the Alaskan and Canadian regions.

Notably, in August, the RMSE of the OnlineML hindcast exceeds that of the Reference hindcast in both the Alaskan and Canadian regions. This is primarily due to the systematic underestimation of SIE by the OnlineML hindcast relative to both the Reference hindcast and the "truth" in these regions (Figure S5). The underlying causes of this systematic underestimation, however, warrant further investigation.

**Comment 24**

Lines 297-299: As the author mentioned, the different performance of these two approaches (OnlineML and OfflineML) comes from the way they are constructed. Therefore, these two methods should be intended for different purposes. Is this comparison appropriate? Maybe rephrasing it would be better.

**Response:** As suggested, we have revised the text as follows (L354-L361 in the manuscript):

The offline approach outperforms the online approach across all regions, primarily because the online correction targets instantaneous model errors (i.e., those on the 15th day of each month). These corrected errors may reemerge through interactions with the other components of the coupled model system, thereby diminishing the overall impact of the online error correction when evaluated using monthly-averaged outputs. In contrast, the offline approach directly adjusts the monthly model outputs, which aligns closely with the evaluation metrics used in this study. Moreover, the offline approach does not need to run the dynamical model and is computationally cheaper than the online approach. However, the online approach not only reduces SIC errors but also propagates corrections through the model integration to the other variables (e.g., sea ice thickness and sea ice drift), ensuring physical consistency between the predicted variables.

> **Comment 25**
>
> Line 302: Does the error correction performance vary with the initialization month (as mentioned above)?

**Response:** As shown in Figures 5, 6, and S1-S4 of the manuscript, the effectiveness of the correction is more sensitive to the target month than to the initialization month. For clarity, we have modified the summary paragraph as follows (L362-L365 in the manuscript):

> In summary, while the error correction performance varies by region and target month, overall, both approaches improve the sea ice prediction. In addition, the offline approach is more efficient than the online approach in reducing the SIE RMSE for both Pan-Arctic and subregions. These conclusions also hold for seasonal predictions initialized in the other seasons. For details, please refer to Figures S1-S4.

**Figures**

> **Comment 26**
>
> Figure 1: The colors in Figure 1 are somewhat confusing (especially the purple and pink, which may cause difficulty for readers in distinguishing them). I recommend to use more distinguishable colors.

**Response:** We thank the reviewer for the comment. As suggested, we have replotted Figure 1 with more distinguishable colors. The revised figure is also presented below as Figure R2.

> **Comment 27**
>
> Figure 2: "Regional domain definitions for Central Arctic, Atlantic, Siberian, Alaskan, Canadian, and Regions based on sea area definitions in Kimmritz et al. (2019)." The "Regions" should be corrected to "regions". I think it seems a bit crude to combine the Bering and the Sea of Okhotsk into the "Pacific Region", is there any literature to support this approach?

**Response:** We thank the reviewer for the comments. We have corrected "Regions" to "regions" and revised the caption of Figure 2 as follows:

> Regional domain definitions for central Arctic, Atlantic, Siberian, Alaskan, Canadian, and Pacific regions are based on sea area definitions in Kimmritz et al. (2019) and are similar to those used in Bushuk et al. (2024). Atlantic region: Greenland, Ice, Norwegian, Barents and Kara Seas; Siberian region: Laptev and East Siberian Seas; Alaskan region: Chukchi and Beaufort Seas; Canadian region: Canadian archipelago, Hudson Bay, Baffin Bay, and Labrador Sea; Pacific region: Bering Sea and the Sea of Okhotsk.

We decided to group the Bering Sea and Sea of Okhotsk as one region mostly due to similar sea ice characteristics in the Bering Sea and Sea of Okhotsk (e.g., strong seasonality) and their geographic locations. In addition, we aimed to reduce the number of the regions and thus the number of sub-figures (e.g., those shown in Figure 6 of the manuscript). Please note that the other regions are very similar to those used in Bushuk et al. (2024).

[Figure]

**Figure R2.** Schema for the online and offline ML-based error correction approaches. The green line represents the "truth". The gray line represents dynamical prediction without error correction. The purple (blue) line represents prediction with online (offline) ML-based error correction. The purple dashed arrows indicate pauses during the prediction production, facilitating correction to the instantaneous model states.

**Comment 28**

Figure 3: Please indicate in the figure caption that this is the result of reanalysis minus the model (as in Figure 4's caption).

**Response:** As suggested, we have modified the caption of Figure 3 as follows:

[revised manuscript text omitted]

---

## Author Comment (AC2)

**Improving Seasonal Arctic Sea Ice Predictions with the Combination of Machine Learning and Earth System Model**

Addressed Comments for Publication to

**The cryosphere**

by

Zikang He, Yiguo Wang, Julien Brajard, Xidong Wang, Zheqi Shen

**Authors' Response to Reviewer #2**

**Comment 1**

The Arctic Ocean is warming at a faster rate than the rest of the planet. Consequently, both the extent and thickness of sea ice have significantly decreased over the past few decades. These changes pose significant challenges to the reliability of seasonal Arctic sea ice predictions. This manuscript aims to enhance the Norwegian Climate Prediction Model (NorCPM) for seasonal Arctic sea ice prediction by integrating machine learning techniques with the Earth system model. Online and offline modules were selected to perform error corrections. The ultimate goal is to improve NorCPM's forecast performance for the marginal ice zone.

I find this study to be both timely and relevant. It aligns well with the scope of the TC Journal. I am inclined to give a positive recommendation. However, I believe there are several issues that need to be addressed before the manuscript can possibly be considered for publication.

**Response:** We very much appreciate that the reviewer found this study relevant and suitable for The Cryosphere (TC). We thank the reviewer for providing insightful comments that helped to significantly improve the manuscript. We carefully addressed each concern and revised the manuscript. Below, we provide our detailed point-by-point responses to the reviewer's comments.

To enhance the legibility of this response letter, the reviewer's comments are typeset in blue boxes. Rephrased or added statements in the revised version of the manuscript are indicated in gray boxes.

**Comment 2**

(1) L158: "MLP excels in function approximation, making it particularly..." Please explain why the MLP (Multilayer Perceptron) model was chosen over the convolutional neural network (CNN). In my opinion, a CNN is better suited for learning spatial neighboring relationships compared to an MLP model. However, using an MLP model to operate on each grid point could lead to abrupt spatial changes in the predicted values.

**Response:** We sincerely appreciate the reviewer's comment. We explained the choice of MLP in the previous version of the manuscript. For clarity, we have revised the manuscript as follows (L183-L188 in the paper):

> The ML architecture used in this study is a multilayer perceptron (MLP), a fully connected neural network well-suited for capturing complex nonlinear relationships in data. MLP offers several advantages, including flexibility in handling diverse input features, efficient training via backpropagation, and strong generalization when properly regularized. Additionally, MLP is computationally more efficient than complex deep learning architectures such as convolutional neural networks (CNNs) and U-Net. It has been successfully applied to error correction in geophysical modeling (e.g., Yang et al., 2023), as it is computationally efficient and requires less training data (Jia et al., 2019; Watson, 2019).

**Table R1.** Number of parameters of the online and offline ML-based error correction models for each ML model.

| | Online ML-based SIC model | Online ML-based SST/SSS model | Offline ML-based SIC model |
|---|---|---|---|
| BatchNorm | 52 | 52 | 20 |
| Dense layer 1 | 840 | 840 | 360 |
| Dense layer 2 | 1830 | 1830 | 1830 |
| Gate layer | 31 | 31 | 31 |
| Dense layer 3 | 840 | 840 | 360 |
| Dense layer 4 | 1830 | 1830 | 1830 |
| Output | 155 | 31 | 31 |
* * *
**Comment 3**

(2) L171: The attention mechanism is a widely used deep learning technique. I suggest that the authors provide a detailed explanation of the attention mechanism employed in this study. Was a temporal attention mechanism or a self-attention mechanism used in this context? Generally, the number of parameters in an attention module is significantly larger than that in an MLP model. Given the volume of data in this study, there is a risk of overfitting. I would like to see the parameter counts for both the MLP model and the attention mechanism presented separately.
* * *
**Response:** In this study, the attention mechanism employed is a gate-based attention layer, which serves to adaptively reweight the input features. Detailed parameters of each machine learning model, including the gate layer, are provided in Table R1 (i.e., Table 2 in the revised manuscript).

To address the potential risk of overfitting, we have incorporated several regularization strategies, such as early stopping, L2 regularization, and dropout. The corresponding descriptions have been added to the manuscript as follows (L200-L208 in the paper):

> The objective function used in this study is the mean squared error (MSE). Additionally, details regarding the number of parameters for each ML model are provided in Table 2. To reduce the risk of overfitting and improve model generalization, the following strategies are implemented:
>
> - **Batch Normalization**: The inputs of each layer are normalized to reduce internal covariate shift, thus promoting training stability and generalization.
>
> - **L2 Regularization**: A penalty is applied to the output layer weights, effectively discouraging over-complex models and reducing the likelihood of overfitting.
>
> - **Early Stopping**: The validation loss is monitored during training and the training is halted once the validation loss curve does not decline, avoiding overfitting due to the training data.
* * *
**Comment 4**

(3) L184: Training a separate model for each month of the test period (2003 to 2022) is not a convincing design. A model trained for each month during the training period should be generalizable to the test period. In my opinion, there is no need to train a distinct model for each month of each year during the test period.

**Response:** We respectively disagree with the reviewer. As shown in Figure R1, the running training results in lower MAE than the fixed-period training (1992 - 2002). Therefore, we adopted running training in this study. For clarity, we have included this comparison into the manuscript as follows (L210-L220 in the paper):

> We adopt a running training strategy, using data from the 11 years preceding the test set to train the ML models. For instance, to develop error correction models for predictions in 2011 (a test set), we train the model using data from 2000 to 2009 and validate it with data from 2010. Similarly, for predictions in 2021, we use data from 2010 to 2019 for training and data from 2020 for validation. This approach ensures that the ML models leverage the most recent data while maintaining a clear separation between training, validation, and test sets. The primary reason for using running training is the pronounced decline trend in Arctic sea ice observed over recent decades, with substantial differences between earlier ice conditions (e.g., the 1980s) and those of recent years (e.g., the 2010s). We also performed sensitivity studies on the length of the running training set (e.g., the most recent 5 years or all years since 1980) and the comparison between the running training and the fixed-period training (1992-2002), which are not shown in the paper. We found that the data from the most recent 11 years leads to the best performance for ML training, and the running training outperforms the fixed-period training.

**Other Comments**

**Comment 5**

The manuscript suffers some unclarities concerning the structure.

**Response:** We have carefully addressed each comment of the three reviewers and revised the manuscript. We believe the manuscript has been improved.

**Comment 6**

(4) L140: This section introduces the limitations of online error correction. However, its placement in the methods description section feels abrupt and lacks strong contextual relevance. I suggest moving this part to the discussion section, where it can be framed as a prospect for future work.

**Response:** Sorry for the confusion. This paragraph describes the post-processing after applying the online error correction approach. For clarity, we have revised the manuscript as follows (L158-L166 in the paper):

> Before restarting the model after applying online error correction, it is essential to ensure that the updated variables remain within physical limits (e.g., SIC between 0% and 100%) and maintain consistency with non-updated variables. If unphysical values or inconsistencies arise, they can lead to model instability. To prevent these issues, we apply a post-processing method specifically designed for NorCPM (Kimmritz et al., 2018):
>
> – If SIC in any thickness category falls below 0% or exceeds 100%, it is set to 0% or 100%, respectively.
>
> – If the total SIC across all thickness categories exceeds 100%, SIC values in each category are proportionally scaled to ensure the total does not surpass 100%.

[Figure]

**Figure R1.** Top row: "true" errors of SIC in the middle of the month based on the analysis increments (i.e., the changes thanks to monthly DA in the reanalysis). Middle row: the errors predicted by the fix-period training online error correction model (1992 - 2002). Bottom row: the errors predicted by the online error correction model. These errors are averaged over the period 2003-2021. Values in the bottom row are the MAE between the "true" and predicted errors across space.

– Sea ice volume in each category is adjusted proportionally to changes in SIC while preserving the ice thickness.

This approach ensures physical constraint and model stability after the error correction.
* * *
**Comment 7**

(5) L165: "The MLP architecture consists of five layers" Please consider presenting this with a diagram for better clarity and algorithm flow.

**Response:** We thank the reviewer for the comment. Our model is relatively simple, and the study involves three different input/output configurations. To avoid potential confusion for readers, we decided not to include a schematic diagram. Instead, we have revised the textual descriptions in the manuscript to clarify the model structures (both ML configuration and the number of parameters) as follows (L189-L208 in the manuscript):
* * *
The entire MLP architecture consists of seven layers:

– **Input layer**: A batch normalization layer (Ioffe, 2017), which helps stabilize and accelerate the training process by normalizing the input features.

– **Second layer**: A dense layer with 60 neurons, using the rectified linear unit (ReLU) activation function.

– **Third layer**: A dense layer with 30 neurons, also employing the ReLU activation function. This layer shares the same structure as the second layer.

– **Fourth layer**: An attention mechanism implemented via a gate layer, which enables the model to focus on important features, thereby enhancing learning efficiency and predictive performance.

– **Fifth layer**: A dense layer with 60 neurons and ReLU activation, mirroring the configuration of the second layer.

– **Sixth layer**: A dense layer with 30 neurons and ReLU activation, identical to the third layer.

– **Output layer**: A dense layer activated by the linear function.

The objective function used in this study is the mean squared error (MSE). Additionally, details regarding the number of parameters for each ML model are provided in Table 2. To reduce the risk of overfitting and improve model generalization, the following strategies are implemented:

– **Batch Normalization**: The inputs of each layer are normalized to reduce internal covariate shift, thus promoting training stability and generalization.

– **L2 Regularization**: A penalty is applied to the output layer weights, effectively discouraging over-complex models and reducing the likelihood of overfitting.

– **Early Stopping**: The validation loss is monitored during training and the training is halted once the validation loss curve does not decline, avoiding overfitting due to the training data.

**Comment 8**

(6) L172: The specific use of the "linear activation function" should be clarified. Did the authors apply an activation mechanism, or was it unnecessary? These details are critical for understanding the implementation.

**Response:** Sorry for the confusion. The output layer is activated by the linear function, which applies no nonlinear transformation to the output. This choice is deliberate, as the task involves regression and requires unbounded continuous output values. Therefore, no activation mechanism (e.g., ReLU or sigmoid) is applied in this layer. We have modified the description as follows (L189-L199 in the manuscript):

> The entire MLP architecture consists of seven layers:
>
> - **Input layer**: A batch normalization layer (Ioffe, 2017), which helps stabilize and accelerate the training process by normalizing the input features.
>
> - **Second layer**: A dense layer with 60 neurons, using the rectified linear unit (ReLU) activation function.
>
> - **Third layer**: A dense layer with 30 neurons, also employing the ReLU activation function. This layer shares the same structure as the second layer.
>
> - **Fourth layer**: An attention mechanism implemented via a gate layer, which enables the model to focus on important features, thereby enhancing learning efficiency and predictive performance.
>
> - **Fifth layer**: A dense layer with 60 neurons and ReLU activation, mirroring the configuration of the second layer.
>
> - **Sixth layer**: A dense layer with 30 neurons and ReLU activation, identical to the third layer.
>
> - **Output layer**: A dense layer activated by the linear function.

**Comment 9**

(7) L175: Please provide a detailed split of the datasets. It is essential to ensure that there is no overlap in time or data between the training set, validation set, and test set to prevent data leakage. Such overlap could lead to an unreliable evaluation of the model's performance on the test set.

**Response:** We agree with the reviewer on the importance of avoiding overlap between the training, validation, and test sets to prevent data leakage. In our approaches, we implement a running training strategy, and there is no overlap between these sets. To enhance clarity, we have revised the relevant paragraph as follows (L210-L220 in the paper):

> We adopt a running training strategy, using data from the 11 years preceding the test set to train the ML models. For instance, to develop error correction models for predictions in 2011 (a test set), we train the model using data from 2000 to 2009 and validate it with data from 2010. Similarly, for predictions in 2021, we use data from 2010 to 2019 for training and data from 2020 for validation. This approach ensures that the ML models leverage the most recent data while maintaining a clear separation between training, validation, and test sets. The primary reason for using running training is the pronounced decline trend in Arctic sea ice

[Figure]

**Figure R2.** Training and validation loss curves of the online error correction models for the test year 2003.

observed over recent decades, with substantial differences between earlier ice conditions (e.g., the 1980s) and those of recent years (e.g., the 2010s). We also performed sensitivity studies on the length of the running training set (e.g., the most recent 5 years or all years since 1980) and the comparison between the running training and the fixed-period training (1992-2002), which are not shown in the paper. We found that the data from the most recent 11 years leads to the best performance for ML training, and the running training outperforms the fixed-period training.

**Comment 10**

(8) L230: It is necessary to evaluate the performance of the MLP model, such as giving specific training set accuracy, validation set accuracy, and test set accuracy, so as to demonstrate the generalization ability of the model and make the subsequent evaluation of specific correction effects more credible.

**Response:** We thank the reviewer for the comment. We have carefully examined the training curves. Considering the large number of ML models trained, we present the results for the online error correction models in the year 2003 as a case study. As shown in Figure R2, the training curves for all twelve months demonstrate satisfactory convergence. Owing to the use of early stopping, the number of training epochs varies among the different months.

[Figure]

**Figure R3.** Top row: "true" errors of SIC in the middle of the month based on the analysis increments (i.e., the changes thanks to monthly DA in the reanalysis). Bottom row: the errors predicted by the online error correction model. These errors are averaged over the period 2003-2021. Values in the bottom row are the MAE between the "true" and predicted errors across space.

**Comment 11**

(9) F3, 4, 7: Please add specific accuracy or error in each subgraph. Present true errors with comma "true errors"

**Response:** As suggested, we have revised these figures by adding a specific accuracy (spatial mean absolute error or $\Delta$RMSE) to each subgraph and introducing a comma for "true" or "truth" in the whole manuscript. Please refer to Figures R3 and R4 and the manuscript.

[Figure]

**Figure R4.** Top row: "true" errors of monthly SIC estimated by the Reference hindcast initialized in July minus the reanalysis. Bottom row: the errors predicted by the offline approach. The errors are averaged over the period 2003-2021. Values in the bottom row are the MAE between the "true" and predicted errors across space.
* * *
**Comment 12**

(10) L261: "Compared with the OnlineML hindcasts, the OfflineML hindcasts have a larger error reduction, particularly in September"; L269: "demonstrates larger error reductions in IIEE than the online approach..."; L273: "The offline approach outperforms the online approach in reducing both RMSE for SIE and IIEE for ice edge, especially in months with higher prediction errors"
The manuscript contains numerous vague expressions. Please provide more specific and concrete details. For example, instead of stating that "the error has been reduced," specify by how much (e.g., "the error has been reduced by xx%"). Including precise values of accuracy is essential for a comprehensive evaluation.

**Response:** We appreciate the reviewer's comment. As suggested, we have revised the manuscript by providing concrete details as follows (L309-L310, L317-L320, L321-L325, L279-L285 and L369-L377 in the manuscript):

[revised manuscript text omitted]

---

## Author Comment (AC3)

**Improving Seasonal Arctic Sea Ice Predictions with the Combination of Machine Learning and Earth System Model**

Addressed Comments for Publication to

**The cryosphere**

by

Zikang He, Yiguo Wang, Julien Brajard, Xidong Wang, Zheqi Shen

**Authors' Response to Reviewer #3**

> ### Comment 1
>
> The manuscript evaluates the performance of machine-learning based error correction models in a coupled earth system model, NorCPM. The evaluation compares both online and offline correction schemes. In this study, the machine learning model is trained and validated against the reanalysis generated from the same coupled earth system model, which is used as the truth. The online scheme is trained to correct instantaneous state wheares the offline scheme is used to correct monthly biases. The manuscript shows improvements using both correction schemes but the offline scheme as a post-processing method beats the online scheme. This work is interesting. However, I recommend that the manuscript should be reconsidered for publication after revision.

**Response:** We very much appreciate that the reviewer found this study interesting. We thank the reviewer for providing insightful comments that have helped to significantly improve the manuscript. We carefully addressed each concern and revised the manuscript. Below, we provide our detailed point-by-point responses to the reviewer's comments. To enhance the legibility of this response letter, all the reviewer's comments are typeset in blue boxes. Rephrased or added sentences in the revised version are indicated in a gray box.

**Major comments:**

> ### Comment 2
>
> 1. In the comparison between the online and offline schemes, the online scheme is applied at 15-th of each month similar to the reanalysis system. However, the manuscript lacks the discussion on the application of the DA increment in the reanalysis system. For example, does the reanalysis system use any incremental analysis update or nudging to provide a continuous correction? The manuscript also lacks the information on the updated ocean and ice state of the reanalysis system. Are they the same as the online correction scheme? For example, there is no mention of SSS in Sect.2.2 but is corrected in the online scheme.

**Response:** We thank the reviewer very much for these comments. For clarity, we have modified the description on the reanalysis dataset used in this study (L120-L134 and L136-L138 in the manuscript):

> The reanalysis is available from 1980 to 2021 with 30 ensemble members. The initial states of the reanalysis on 15 January 1980 are taken from a NorESM ensemble run integrated from 1850 to 1980 with CMIP5 historical forcings. In this reanalysis, NorCPM assimilates monthly anomalies of SST, SIC, and subsurface hydrographic profile data in the middle of each month.
>
> From 1980 to 2002, the climatology used for anomaly-field assimilation is defined over the period 1980–2010. SST and SIC observations are from HadISST2 (Titchner and Rayner, 2014) and subsurface hydrographic profile data from EN4.2.1 (Good et al., 2013). The assimilation process contains two steps addressed in Kimmritz et al. (2019): firstly, hydrographic DA updates the ocean state (Wang et al., 2017). Subsequently, SST and SIC DA occur and update the sea ice and ocean states within the ocean mixed layer. From 2003 to 2021, the climatology utilized for anomaly-field assimilation is defined from 1982 to 2016. SST and SIC observations are from OISST (Reynolds et al., 2007) and subsurface hydrographic profile data from EN4.2.1 (Good et al., 2013). Strong-coupled DA is performed to simultaneously update the sea ice and ocean states in a single step. After each assimilation step, a post-processing step is used to ensure the physical consistency of state variables. For example, the volume of each sea ice category is proportionally adjusted based on the updated SIC

(Kimmritz et al., 2018, 2019). The other model components, such as the atmosphere and land, are dynamically adjusted through the coupler during model integration between two assimilation steps.

The online error correction approach is built from the analysis increment of the reanalysis introduced in section 2.2 (Brajard et al., 2021; Gregory et al., 2024) and sequentially applied to update the instantaneous model state in the middle of each month during prediction simulation (purple line in Figure 1), which is similar to the reanalysis system (section 2.2).

**Comment 3**

These information can be useful because, in a perfect scenario, if the online correction scheme can give the same increment as the analysis increment, the online correction scheme should be able to recover the reanalysis deemed as truth here. This, of course, cannot be the case in reality, but can be useful for discussing the sources of delta RMSE. For example, lack of spatial correlation due to training on individual grid points, different strategies for correction/increment applications, or the possibility that instantaneous random errors can be averaged out so that the ML only learns systematic biases during the training processes due to the long-term data being used.

**Response:** We agree with the reviewer on this comment. For clarity, we have added discussions on the remain/unpredicted error into the manuscript (L413-L422 in the manuscript):

Furthermore, the current ML model (MLP) is trained independently at each grid point and thus cannot capture spatial correlations. This limits its ability to correct spatially coherent errors, particularly in regions where NorCPM already performs well and only subtle adjustments are needed. As a result, the hybrid model often struggles to reproduce the reanalysis, which are treated as the "truth" in this study. While it is unrealistic to expect the model to perfectly replicate analysis increments, the discrepancy is closely related to the ML-based model's learning capacity and the nature of the underlying errors. Possible contributing factors include: (1) the lack of spatial dependencies due to pointwise training and (2) the tendency of models trained on long-term data to learn systematic biases rather than instantaneous random errors, the latter of which tend to be averaged out over time. Therefore, there is still room to improve the ML-based error correction framework. Future studies could explore spatially-aware architectures, such as CNNs and U-Net, and incorporate additional predictors to capture complex error structures and enhance correction performance (Palerme et al., 2024).

**Comment 4**

2. The definition of error needs to be reformulated. The analysis increment is the differences between the analysis and forecast, which is equivalent to the differences between the analysis and forecast error, xa - xf = ea - ef. Even if we take ea = 0, the increment is the negative error of the xf. Therefore, if Eq.(2) is the estimated model error, Eq.(3) means that an error is added to the model forecast. In fact, the error should be removed. The authors may want to add a negative sign to Eq.(2). The same logic can be applied to Sect.3.1 where I believe the negative error, instead of the actual error is presented.

**Response:** We agree with the reviewer that the definitions and equations in sections 2.3, as well as Figures 3 and 4 in section 3.1, are confusing. For clarity, we have revised the relevant paragraph in section 2.3 as follows (L144-L152 in the manuscript):

> The online approach is to emulate the difference between the forecast and the analysis $\mathbf{x}_k^f - \mathbf{x}_k^a$, which corresponds to the opposite of the analysis increment in DA. The error prediction model can be expressed as:
>
> $$\varepsilon = \mathcal{M}_{\mathrm{e}}(\mathbf{x}^f), \tag{1}$$
>
> where $\mathcal{M}_{\mathrm{e}}$ represents the data-driven model taking the instantaneous model state $\mathbf{x}^f$ as input and $\varepsilon$ represents the predicted model error.
> The hybrid model, incorporating the dynamic model and the online error correction model, can be expressed as follows:
>
> $$\mathbf{x}_l^h = \mathcal{M}(\mathbf{x}_{l-1}^h) - \mathcal{M}_{\mathrm{e}}(\mathcal{M}(\mathbf{x}_{l-1}^h)), \tag{2}$$
>
> where $\mathbf{x}_l^h$ represents the error-corrected instantaneous model state at $t_l$ during the prediction.
> We aim to correct SIC, SST, and SSS errors in the ice-covered area, which are directly associated with the sea ice condition.

To be consistent, we have revised Eq. (3) as follows (L150 in the manuscript):

> $$\mathbf{x}_l^h = \mathcal{M}(\mathbf{x}_{l-1}^h) - \mathcal{M}_{\mathrm{e}}(\mathcal{M}(\mathbf{x}_{l-1}^h)), \tag{3}$$

Accordingly, we have revised Figures 3 and 4 of the manuscript (Figures R1 and R2).

> **Comment 5**
>
> 3. As one of the selling point of this manuscript is the use of fully coupled ESM, can the authors provide some analysis and discussions on the ocean state as well such that one can get a better physical intuition of the results?

**Response:** We thank the reviewer for the valuable suggestion. We have analyzed SST and SSS. Taking the July initialization as an example, which is consistent with Figure 7 of the manuscript, we found that both SST and SSS exhibit clear improvements, particularly along the ice-edge region (Figures R3 and R4). We have added discussions into the manuscript (L398-L405 in the manuscript):

> By comparing the two error correction approaches, we find that the offline approach yields smaller errors than the online approach. The online error correction approach corrects instantaneous model errors only on the 15th day of the month, and the effect of this correction gradually weakens during model integration due to the accumulation of errors in the other model components. Consequently, the impact of the correction becomes less evident when computing monthly-averaged outputs. Nevertheless, the online error correction can reduce errors in SST and SSS (Figures S7 and S8). Moreover, the online correction approach maintains better physical consistency among the predicted variables through dynamical model integration. The offline error correction approach directly corrects the model outputs without requiring model integration. As a result, it is

[Figure]

**Figure R1.** Top row: "true" errors of SIC in the middle of the month based on the analysis increments (i.e., the changes thanks to monthly DA in the reanalysis). Bottom row: the errors predicted by the online error correction model. These errors are averaged over the period 2003-2021. Values in the bottom row are the MAE between the "true" and predicted errors across space.

computationally more efficient and easier to integrate into operational sea ice prediction systems than the online approach.

**Minor comments:**

**Comment 6**

1. L28: perhaps reads better with "transitioning to DA methods to...."

**Response:** We have revised the text as follows (L31-L33 in the manuscript):

Simultaneously, many prediction centers are transitioning to use DA methods to mitigate uncertainties in initial conditions (Wang et al., 2013; Vitart et al., 2017; Blockley and Peterson, 2018; Kimmritz et al., 2019; Wang et al., 2019; Bushuk et al., 2024).

[Figure]

**Figure R2.** Top row: "true" errors of monthly SIC estimated by the Reference hindcast initialized in July minus the reanalysis. Bottom row: the errors predicted by the offline approach. The errors are averaged over the period 2003-2021. Values in the bottom row are the MAE between the "true" and predicted errors across space.
* * *
**Comment 7**

2. L73: Does NorCPm use DEnKF instead of stochastic EnKF? Would it be more informative to cite SAKOV, P. and OKE, P.R. (2008), A deterministic formulation of the ensemble Kalman filter: an alternative to ensemble square root filters.

**Response:** The reviewer is right. We do use DEnKF (a deterministic version of EnKF). For clarity, we have revised the manuscript as follows (L77-L78 in the manuscript):

It combines the Norwegian Earth System Model version 1 (NorESM1, Bentsen et al., 2013) and a deterministic formulation of an advanced flow-dependent DA method named ensemble Kalman filter (EnKF, Sakov and Oke, 2008).
* * *
**Comment 8**

3. L100: I'm not sure EnKF used here actually provide a spatiotemporal estimate as normally filtering only provide spatial correlation in their error. Perhaps it is better to say "time-dependent spatial error estimate"?

[Figure]

**Figure R3.** Differences between SST RMSE of the Reference and OnlineML hindcasts initialized from July. Warmer (colder) colors indicate that the OnlineML hindcast performs better (worse).

**Response:** We thank the reviewer for the comment. For clarity, we have modified the text as follows (L103-L105 in the manuscript):

> The EnKF accounts for uncertainties in initial conditions to generate ensemble predictions, which evolve in time and provide time- and space-dependent error estimates.

**Comment 9**

4. L140-143: It should be explicitly said that the same post-processing of NorCPM is used in the online correction scheme.

**Response:** We have revised the manuscript by providing explicit information on the post-processing used in the study (L158-L166 in the manuscript):

> Before restarting the model after applying online error correction, it is essential to ensure that the updated variables remain within physical limits (e.g., SIC between 0% and 100%) and maintain consistency with non-updated variables. If unphysical values or inconsistencies arise, they can lead to model instability. To

[Figure]

**Figure R4.** Differences between SSS RMSE of the Reference and OnlineML hindcasts initialized from July. Warmer (colder) colors indicate that the OnlineML hindcast performs better (worse).

prevent these issues, we apply a post-processing method specifically designed for NorCPM (Kimmritz et al., 2018):

- If SIC in any thickness category falls below 0% or exceeds 100%, it is set to 0% or 100%, respectively.

- If the total SIC across all thickness categories exceeds 100%, SIC values in each category are proportionally scaled to ensure the total does not surpass 100%.

- Sea ice volume in each category is adjusted proportionally to changes in SIC while preserving the ice thickness.

This approach ensures physical plausibility and model stability after error correction.

**Comment 10**

5. Sect.2.4: Is post-processing applied to output from offline schemes for physical consistency when comparing online and offline schemes?

**Response:** We thank the reviewer for the question. There was a minor treatment for unphysical values in the offline scheme. For clarity, we have modified the text as follows (L169-L171 in the manuscript):

The input features are monthly SST, SSS, total SIC, and latitude. The output feature is the error in the monthly SIC. The predicted error is subtracted from the monthly SIC. If the updated monthly SIC falls below 0% or exceeds 100%, it is set to 0% or 100%, respectively.

**Comment 11**

6. Sect. 2.5: what is the objective function being used here? Is it RMSE?

**Response:** The objective function used in this study is the mean squared error (MSE). We have added the following text in the manuscript as follows (L200 in the manuscript):

The objective function used in this study is the mean squared error (MSE).

**Comment 12**

7. L193: Why is the reference configuration performed from 1991 - 2002 which is not performed for the online experiment?

**Response:** We thank the reviewer for the comment. There was a typographical error; the data actually start from 1992. In our approach, we employ a rolling training strategy, using data from the 11 years preceding each test year to train the ML models. This design ensures that no "future" information is used during training or validation. Since our test period spans 2003–2021, the first test case in 2003 requires training and validation data from 1992 to 2002. Consequently, this period could not be included in the test set. For further details, please refer to the relevant paragraph in the manuscript as follow (L210–L220 in the manuscript):

We adopt a running training strategy, using data from the 11 years preceding the test set to train the ML models. For instance, to develop error correction models for predictions in 2011 (a test set), we train the model using data from 2000 to 2009 and validate it with data from 2010. Similarly, for predictions in 2021, we use data from 2010 to 2019 for training and data from 2020 for validation. This approach ensures that the ML models leverage the most recent data while maintaining a clear separation between training, validation, and test sets. The primary reason for using running training is the pronounced decline trend in Arctic sea ice observed over recent decades, with substantial differences between earlier ice conditions (e.g., the 1980s) and those of recent years (e.g., the 2010s). We also performed sensitivity studies on the length of the running training set (e.g., the most recent 5 years or all years since 1980) and the comparison between the running training and the fixed-period training (1992-2002), which are not shown in the paper. We found that the data from the most recent 11 years leads to the best performance for ML training, and the running training outperforms the fixed-period training.

**Comment 13**

8. L207: What is an areal sum of grid points? Is it an area-weighted sum of the SIC over all grid points with SIC >= 15%?

**Response:** Sorry for the confusion. For clarity, we have revised the manuscript as follows (L247-L249 in the manuscript):

> In this study, the SIE is defined as the total area of all grid points within the region of interest where SIC $\geq$ 15%. SIE is calculated for each ensemble member, and we evaluate the ensemble mean by averaging SIE across all ensemble members.

**Comment 14**

9. L226-229: What does "10 data points from the 10 ensemble members" mean? Do you mean selecting one data point from each of the 10 ensemble members leading to 10 data points in total, or do you mean selecting 10 data points from each of the 10 ensemble members leading to 100 data points in total? Based on the RMSE in Eq.(4), the RMSE is calculated over time, this means that one RMSE will be obtained for each grid point. How can a single RMSE over both time and space be obtained? Are there any results for the uncertainty of the RMSE in this manuscript?

**Response:** Sorry for the confusion. We aimed to estimate the RMSE uncertainties related to the ensemble mean. We resampled with replacement 10 data from the ensemble, ensuring the data size of 10 is equal to the ensemble size. As Eq. (5), we compute RMSE for both SIC and SIE over time and not over space (e.g., Figures 5-7 in the manuscript). For clarity, we have revised the manuscript as follows (L271-L275 in the manuscript):

> To estimate the uncertainties in an RMSE value arising from the small ensemble size, we employ the bootstrap method. Specifically, we randomly sample 10 ensemble members with replacement from the ensemble, compute the ensemble mean, and then calculate the RMSE (for either SIC or SIE) based on this resampled data. This process is repeated 10,000 times, producing a distribution of 10,000 RMSE values. The standard deviation of this distribution is then used to quantify the uncertainties associated with the RMSE value.

In Figures 5b-c of the manuscript, we used the uncertainties to perform the significant test. The $\Delta$RMSE that does not pass the 95% significance test was marked with a black dot. In Figure 6 of the manuscript, the uncertainties of SIE RMSE were shown as error bars. In Figure 7 in the manuscript, the RMSE differences that are not statistically significant were plotted as white colors.

**Comment 15**

10. L232: "...in predicting..."

**Response:** As suggested, we have revised the text as follows (L278 in the manuscript):

[revised manuscript text omitted]

---

## Author Response (AR4)

**Correcting Errors in Seasonal Arctic Sea Ice Prediction of Earth System Model with Machine Learning**

Addressed Comments for Publication to

**The cryosphere**

by

Zikang He, Yiguo Wang, Julien Brajard, Xidong Wang, Zheqi Shen

**Authors' Response to Reviewer #1**

> ### Comment 1
>
> Overall, the authors responded comprehensively and satisfactorily to my concerns. They substantially expanded the discussion regarding the online vs. offline approach comparisons, the ML correction vs. pure data-driven comparison, and potential future improvements. I also commend that the title and abstract have been improved to be clearer.

**Response:** We sincerely appreciate the reviewer's thoughtful and encouraging comments and are pleased to hear that the revisions have adequately addressed the reviewer's concerns.

Below, we provide detailed, point-by-point responses to further comments of the reviewer. For clarity, reviewer comments are shown in blue boxes, and the corresponding revisions in the manuscript are highlighted in gray boxes.

> ### Comment 2
>
> One thing I'm still a little confused about is that the authors mention that only considering latitude information for features, and not using longitude, is taking into account that 0 and 360 can't portray that it's actually the same location. So why not convert to a polar stereographic projection grid? That way (lon, lat) is converted to (x, y) coordinates, which can reasonably characterize the actual distance.

**Response:** We fully agree with the reviewer that using a polar stereographic projection to convert (lon, lat) to (x, y) coordinates can better capture spatial relationships, especially around the 0°/360° discontinuity. We will consider adopting this approach in future studies to further improve our model's spatial feature representation.

> ### Comment 3
>
> Additionally, I suggest that the authors carefully revise the manuscript one final time before final resubmission. The language is not necessarily incorrect, but I found a lot of locations where it could be more succinct and precise. A (non-comprehensive) list of edits is below:

**Response:** We sincerely thank the reviewer for the valuable feedback regarding the language of the manuscript. In response, we have thoroughly revised the text to enhance clarity, readability, and conciseness. We have carefully addressed all of the reviewer's specific suggestions and have also conducted a comprehensive final review of the manuscript. Additionally, we enlisted the help of Marianne Williams-Kerslake, a native English speaker and colleague at NERSC, who provided a detailed review and refinement of the English language throughout the manuscript.

> ### Comment 4
>
> Line 26: Not sure if I missed something, but I did not find any statement like "must further improve ..." in Bushuk et al. (2024). Perhaps the following expression is closer to the meaning of the original: the improvement on initialization and model resolution is expected to facilitate the predictions.

**Response:** As suggested, we have adjusted the sentence to better reflect the original meaning as follows (L26-L27 in the manuscript):

Bushuk et al. (2024) also suggested that improving initialization and model resolution is expected to facilitate predictions.

**Comment 5**

Lines 42-55: I didn't understand the necessity of splitting it into two paragraphs because it's all about the implications of online correction.

**Response:** As suggested, we have merged them into a single paragraph to improve the coherence and readability of the text as follows (L42-L53 in the manuscript):

In the context of the online error correction, ML is applied to correct errors in the instantaneous model state (i.e., initial conditions for the following model integration) and sequentially applied to update the instantaneous model state during simulation (e.g., Brajard et al., 2021), referring to an ML-dynamical hybrid model (purple line in Figure 1). Such online error correction approaches have been investigated in both an idealized framework (e.g., Watson, 2019; Brajard et al., 2021) and real applications (e.g., Watt-Meyer et al., 2021). Watson (2019) examined the tendency error correction approach in the Lorenz 96 model. Brajard et al. (2021) explored the resolvent error correction approach in the two-scale Lorenz model as well as in a low-order coupled ocean-atmosphere model called the Modular Arbitrary-Order Ocean-Atmosphere Model (MAOOAM, De Cruz et al., 2016). Watt-Meyer et al. (2021) demonstrated that the online error correction can improve the short-term forecasting skill and accuracy of precipitation simulation while the dynamical model can run indefinitely without numerical instabilities arising. Gregory et al. (2024) applied ML to correct sea ice errors in an ocean-ice coupled model and demonstrated that ML can effectively reduce sea ice bias in a 5-year simulation. So far, the ML-based online error correction method has not been tested for seasonal sea ice prediction in an Earth system model.

**Comment 6**

Lines 63 and 65: "In this study" repeated in two consecutive sentences could be rephrased. Or, remove the sentence in lines 54-55 and 63-64, as lines 65-67 already express the same meaning.

**Response:** We sincerely thank the reviewer for the valuable suggestions. As suggested, we have removed these sentences.

**Comment 7**

Line 420: This sentence, "Therefore, there is still ... framework." could be removed as the previous paragraph has a similar statement (line 406).

**Response:** As suggested, we have removed this sentence.

**References**

Brajard, J., Carrassi, A., Bocquet, M., and Bertino, L.: Combining data assimilation and machine learning to infer unresolved scale parametrization, Philosophical Transactions of the Royal Society A, 379, 20200 086, 2021.

Bushuk, M., Ali, S., Bailey, D. A., Bao, Q., Batté, L., Bhatt, U. S., Blanchard-Wrigglesworth, E., Blockley, E., Cawley, G., Chi, J., et al.: Predicting September Arctic Sea Ice: A Multi-Model Seasonal Skill Comparison, Bulletin of the American Meteorological Society, 2024.

De Cruz, L., Demaeyer, J., and Vannitsem, S.: The modular arbitrary-order ocean-atmosphere model: MAOOAM v1. 0, Geoscientific Model Development, 9, 2793–2808, 2016.

Gregory, W., Bushuk, M., Zhang, Y., Adcroft, A., and Zanna, L.: Machine learning for online sea ice bias correction within global ice-ocean simulations, Geophysical Research Letters, 51, e2023GL106 776, 2024.

Watson, P. A.: Applying machine learning to improve simulations of a chaotic dynamical system using empirical error correction, Journal of Advances in Modeling Earth Systems, 11, 1402–1417, 2019.

Watt-Meyer, O., Brenowitz, N. D., Clark, S. K., Henn, B., Kwa, A., McGibbon, J., Perkins, W. A., and Bretherton, C. S.: Correcting weather and climate models by machine learning nudged historical simulations, Geophysical Research Letters, 48, e2021GL092 555, 2021.

---

## Author Response (AR5)

Responses to Editor's Comments for Manuscript

**Correcting Errors in Seasonal Arctic Sea Ice Prediction of Earth System Model with Machine Learning**

Addressed Comments for Publication to

by

Zikang He, Yiguo Wang, Julien Brajard, Xidong Wang, Zheqi Shen

**Authors' Response to Editor**

**Comment 1**

Dear Authors,
Thank you for uploading the revised manuscript. I am pleased to accept it for publication in TC. Please ensure the text format, figures, and tables comply with the TC standard (e.g. line 57 "On the other hand" is continuous with the previous sentence rather than starting a new paragraph). Congratulations.
Best regards, Bin Cheng

**Response:** We sincerely appreciate the editor's acceptance of our manuscript. As suggested, we have removed "On the other hand" in line 57.